# Large-scale exome analyses reveal new rare variant contributions in amyotrophic lateral sclerosis

Amyotrophic lateral sclerosis (ALS) is a heritable disorder where rare variants with low-to-moderate penetrance are thought to dominate genetic risk. To identify such rare variants, we harmonized and analyzed exome data from 22 cohorts, totaling 17,919 individuals with ALS and 200,703 controls across discovery and replication phases. Rare variant analyses identified several new risk genes, with replication confirming association of *YKT6* and supporting *HTR3C*, *GBGT1* and *KNTC1*. We also provide strong, independent validation for genes with limited previous evidence: *ARPP21*, *DNAJC7* and *CFAP410*. Notably, in *ARPP21*, we identified a new high-effect variant (p.P747L) and confirmed that p.P563L is an ALS-associated variant leading to an aggressive disease course. Beyond new discoveries, our analyses largely recapitulated the known genetic architecture of ALS, identifying risk variants in over 20% of cases and supporting a cumulative oligogenic risk model. These findings highlight new translational targets and show that rare variant analyses capture substantially more genetic risk than common variant genome-wide association studies.

ALS is a fatal neurodegenerative disease with a substantial genetic component[1-3]. Despite the discovery of several ALS genes, the genetic etiology remains elusive for most patients, with previous work indicating that a substantial portion of genetic risk for ALS is mediated by rare variants[4]. Identifying new ALS genes is key to furthering our understanding of the disease and may provide direct potential therapeutic targets, as evidenced by the recent approval of Tofersen for *SOD1* mutation-positive ALS. The advent of next generation sequencing has opened the way for population-scale, genome-wide studies of both familial and sporadic ALS cases, which has already led to the identification of several ALS genes, including, among others, *TBK1*, *NEK1* and *KIF5A*[5-9]. In continuing the search for ALS genes, sequencing data from tens of thousands of people are required to enable detection of ultrarare and low-to-modest impact variants. To this end, we assembled and harmonized data from 22 cohorts to generate the largest ALS exome sequencing dataset to date. This provided a discovery cohort encompassing 13,138 cases and 69,775 controls and an independent replication set of 4,781 cases and 130,928 controls. We processed and harmonized all samples uniformly, including realignment to the GRCh38

reference genome and joint variant calling, which we show to be crucial to eliminate structural biases. Through comprehensive single-variant and ultrarare variant (URV) burden analyses, we identified several new candidate variants, genes and genesets, thereby substantially expanding our understanding of the contribution of rare variants to ALS.

## Results

### Building a harmonized ALS exome dataset for rare variant analysis

To identify rare coding variants involved in ALS, we harmonized 18 whole-exome (WXS) and whole-genome (WGS) sequencing datasets into a discovery cohort totaling 94,545 people. All data were realigned uniformly to GRCh38 and called jointly using the functional equivalence pipeline[10,11], substantially reducing technical variation (Supplementary Fig. 1a). Moreover, the distributions of exome-wide URV counts were aligned between ancestry-matched WGS (Project MinE) and WXS (UK Biobank) samples, indicating that sequencing technologies were comparable after joint processing and quality control (Supplementary Fig. 1b). Following strict filtering, the final dataset

✉e-mail: K.P.Kenna@umcutrecht.nl; J.H.Veldink@umcutrecht.nl

comprised 13,138 unrelated cases and 69,775 controls of predominantly European ancestry, with 5,207,138 variants (2,367,861 predicted moderate or high impact; Supplementary Figs. 2–4).

### Rare single-variant analyses identify five new risk variants and largely recapitulate known rare variant architecture of ALS

We conducted single-variant analyses of 272,925 rare variants that fell within our testable minor allele frequency (MAF) range ($5 \times 10^{-5} <$ MAF $< 0.05$) while also satisfying variant effect prediction criteria of either moderate- (missense mutations, in-frame deletions and untranslated region (UTR) truncations) or high-impact (nonsense, splice acceptor/donor and frameshift mutations) annotations. For each variant, we used Firth's logistic regression to test for an association between ALS status and minor allele count (MAC), adjusting for sex, ten principal components (PCs) and the total number of rare synonymous variants in each person[12,13]. The resulting test statistics showed no systematic inflation, indicating no residual confounding ($\lambda_{1,000} = 1.01$), and significant variants passed subsequent validation and sensitivity analyses (Extended Data Fig. 1a,b).

We identified 15 exome-wide significant variants across 11 distinct genes ($P < 1.83 \times 10^{-7}$; Fig. 1a, Table 1, Extended Data Table 1, Extended Data Figs. 1 and 2 and Supplementary Data 1), for all of which the minor allele was associated with increased ALS risk (Fig. 1c). Among the 15 associated variants, 10 were located in genes previously shown to be related to ALS: *SOD1*, *CFAP410*, *NEK1*, *KIF5A*, *FUS* and *TBK1* (Fig. 1a and Extended Data Table 1). The remaining five have not been reported previously in ALS (Table 1). These encompass intermediate frequency variants with modest effect size, including *HTR3C* p.T186A (odds ratio (OR) = 3.41, $P = 1.87 \times 10^{-8}$) and *YKT6* p.Y64C (OR = 2.84, $P = 9.08 \times 10^{-8}$) as well as rare variants with high effect size, including *GBGT1* p.R152L (OR = 26.9, $P = 1.68 \times 10^{-10}$), *CAPN2* p.I530V (OR = 25.3, $P = 3.66 \times 10^{-9}$), and *KNTC1* p.W287R (OR = 27.7, $P = 1.07 \times 10^{-7}$).

We also performed a targeted analysis of variants within 51 ALS-linked genes curated by the ALS Gene Curation Expert Panel (GCEP)[14]. To ensure the inclusion of the full set of GCEP-curated genes, we did not apply the per-supercohort call-rate filter for this analysis, allowing for the assessment of genes exhibiting subpar call-rates in certain subcohorts. This identified eight additional variants across six genes ($P < 3.20 \times 10^{-5}$; Fig. 1b,c, Extended Data Table 1 and Supplementary Data 2), including variants in genes that were not detected in the exome-wide analysis (*ARPP21*, *ANXA11*, *UBQLN2* and *TARDBP*). For all identified variants, the minor allele was associated with increased ALS risk (Fig. 1c and Extended Data Table 1). We provide independent evidence for two rare variants in *ARPP21* (p.P563L and p.P747L)—a gene that is currently considered as having limited evidence according to GCEP (p.P563L: OR = 44.8, $P = 2.55 \times 10^{-10}$; p.P747L: OR = 75.8, $P = 1.45 \times 10^{-6}$) (Fig. 1b,c). Of note, the *ARPP21* p.P563L variant had subpar call-rates in some exome cohorts. However, even when restricting the analysis to cohorts meeting stringent call-rate thresholds, the association remained exome-wide significant with a similar odds ratio ($P = 1.09 \times 10^{-8}$, OR = 38.1; Extended Data Fig. 1c,d).

Principal component analysis (PCA) suggested a mixed pattern of geographical distribution for carriers of the identified variants (Supplementary Fig. 5). For some variants, we observed that carriers exhibited relatively tight clustering in PCA space. This was observed for both well-established ALS variants such as *UBQLN2* p.P509S (Sweden) and *SOD1* p.A5V (USA), as well for the new *CAPN2* p.I540V variant (the Netherlands). Conversely, other variants were distributed more uniformly across patient populations (for example, *YKT6* p.Y64C and *ARPP21* p.P563L). In silico pathogenicity prediction tools also yielded varying annotations for both previously established and new ALS-associated variants (Supplementary Table 1). Nonetheless, we observed that *YKT6* p.Y64C was consistently predicted as damaging by all predictors, and *KNTC1* p.W287R was predicted as damaging by all but SIFT.

### Ultrarare burden analyses identify new ALS-associated genes

To detect associations among URVs (five or fewer carriers), we performed burden tests using Firth's logistic regression to evaluate their cumulative effects. URVs were aggregated across several functional units, including genes and protein domains. To enrich for potentially pathogenic variants, we used four filtering strategies based on two criteria: (1) variant frequency—either all URVs or singleton-only variants; (2) variant impact—either only high-impact variants or both high- and moderate-impact variants. Tests across these filtering strategies were combined using the ACAT omnibus test[15]. We observed no evidence of genomic inflation in any of the analyses performed (gene $\lambda_{1,000} = 1.011$, domain $\lambda_{1,000} = 1.006$; Extended Data Fig. 3a), and all presented genes passed subsequent sensitivity analyses (Extended Data Fig. 3e–g).

URV gene burden analyses across 17,324 protein-coding genes identified eight genes that reached exome-wide significance ($P < 2.89 \times 10^{-6}$) (Fig. 2a, Table 2, Extended Data Table 2, Extended Data Figs. 2 and 3, Supplementary Data 3 and 4 and Supplementary Fig. 6). Among these were four established ALS genes: *SOD1* ($P < 1 \times 10^{-16}$), *TBK1* ($P < 1 \times 10^{-16}$), *NEK1* ($P = 6.49 \times 10^{-13}$) and *TARDBP* ($P = 5.02 \times 10^{-8}$) (Extended Data Table 2). Furthermore, we identified *DNAJC7* ($P = 8.77 \times 10^{-8}$), which is currently classified as having limited evidence (ClinGen gene curation panel[14]), and here reaches exome-wide significance for the first time in an exome-wide discovery analysis. New candidate genes included *TTC3* ($P = 4.16 \times 10^{-7}$), *UNC13C* ($P = 2.80 \times 10^{-7}$) and *KIF4A* ($P = 1.62 \times 10^{-6}$), in all of which higher URV burden increased risk of ALS (Table 2 and Extended Data Fig. 3b). A targeted analysis among the 51 ALS-linked genes classified by GCEP also revealed a significant association for *OPTN* ($P = 1.56 \times 10^{-5}$), which is classified by GCEP as a definitive ALS gene (Fig. 2b and Extended Data Table 2).

The URV domain analyses across 65,071 domains identified three partially overlapping domains in *TBK1* (protein kinase, kinase-like and CCD1 domains), one domain in *SOD1* (SOD_Cu/Zn_BS domain) and one domain in *VCP* (CDC48 domain 2-like domain) at exome-wide significance ($P < 7.68 \times 10^{-7}$; Fig. 2c, Extended Data Fig. 3c and Supplementary Data 5 and 6). Unlike *SOD1* and *TBK1*, *VCP* did not reach significance in the whole-gene analysis ($P_{gene} = 8.09 \times 10^{-3}$), suggesting that the CDC48 domain 2-like region harbors the primary association signal with a markedly stronger effect ($P_{domain} = 2.16 \times 10^{-7}$). This domain constitutes the second subdomain of the N-terminal domain, in which most known pathogenic mutations are concentrated[16].

Across burden analyses, ORs were generally similar when including all URVs compared to including singletons only, with the notable exceptions of *NEK1* and *KIF4A*, which showed markedly higher ORs in the singleton-only analyses (Extended Data Fig. 3b–d). The observed associations were driven primarily by moderate-impact variants: *NEK1* and *TBK1* were the only genes showing a significant signal when analyses were restricted to high-impact variants (Extended Data Fig. 3k–m), although some signal among high-impact variants was observed for *DNAJC7* and *OPTN*. Single nucleotide variants (SNVs) were the primary drivers of the associations, with insertions/deletions (INDELs) contributing substantially to the association *P* values only for *NEK1* and *DNAJC7* (Extended Data Fig. 3h–j). For *UNC13C, TTC3* and *OPTN*, we identified a small subset of people carrying two URVs, whereas for *TBK1* and *NEK1*, we found people with both a URV and a more common (0.01 < MAF < 0.05) risk variant (p.V464A and p.R261H, respectively). No increased risk was observed in these cases, although this may be due to the low number of co-occurrences (Extended Data Fig. 4a).

### Assessing geneset burden and variant co-occurrence

We performed URV geneset burden analyses across 11,777 Gene Ontology (GO), Kyoto Encyclopedia of Genes and Genomes (KEGG) and Reactome genesets from the Molecular Signatures Database (MSigDB v.7.5)[17], using the same procedure as the single gene analyses ($\lambda_{1,000} = 1.006$, Supplementary Data 7 and 8). After excluding genesets driven solely by one highly significant gene, two genesets remained significant:

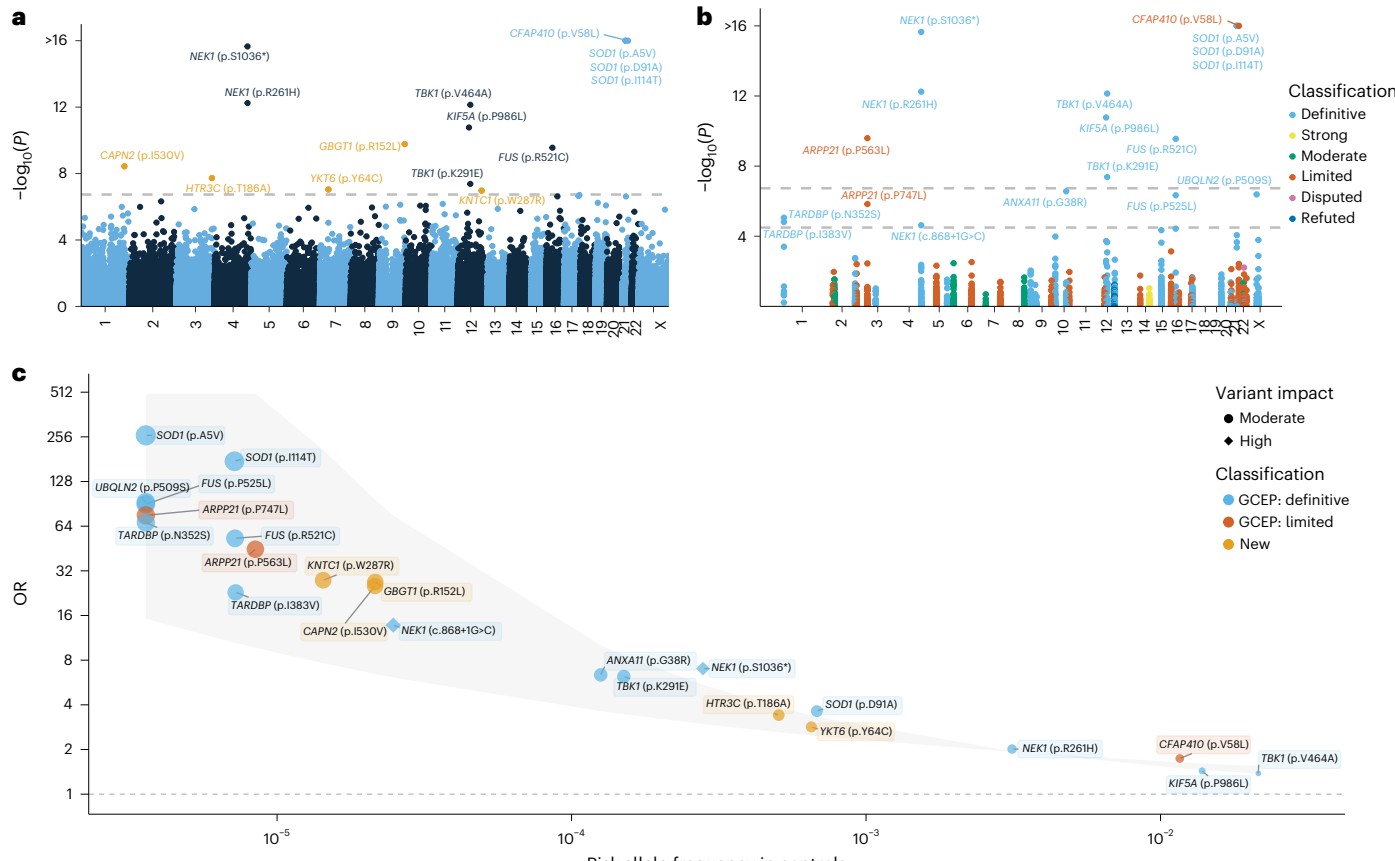

**Fig. 1 | Rare single-variant analyses. a**, *y* axis: exome-wide single variant associations estimated using Firth's logistic regression with profile penalized likelihood CIs ($-\log_{10}(P)$); *x* axis: genomic coordinates (GRCh38). Dashed line: exome-wide significance threshold ($P < 1.83 \times 10^{-7}$). New variants are highlighted in orange. **b**, Rare single-variant analyses among ALS-linked genes curated by the ALS GCEP. *y* axis: single-variant associations estimated using Firth's logistic regression with profile penalized likelihood CIs ($-\log_{10}(P)$); *x* axis: genomic coordinates (GRCh38). Variants are colored by the clinical validity classification

as curated by the ALS GCEP. Lower dashed line: significance threshold across variants in ALS-linked genes ($P < 3.20 \times 10^{-5}$); upper dashed line: exome-wide significance threshold as presented in **a**. **c**, ORs (*y* axis) and 95% CIs (gray shaded area) plotted against the risk allele frequency in controls (*x* axis) for significant variants identified in either the exome-wide or GCEP analysis. For variants where the control risk allele frequency was 0, it was set to half the lowest nonzero risk allele frequency observed in the control group. *P* values are two-tailed and are presented uncorrected for multiple testing.

'GOBP: regulation of mRNA splicing via spliceosome' (GO:0048024, 96 genes, $P = 2.97 \times 10^{-7}$) and its parent term 'GOBP: regulation of RNA splicing' (GO:0043484, 142 genes, $P = 3.50 \times 10^{-6}$) (Fig. 2d and Extended Data Fig. 3d). As 'regulation of mRNA splicing via spliceosome' is a subset of 'regulation of RNA splicing,' we performed a conditional analysis to assess its independent contribution. This revealed that residual signal remains in 'regulation of mRNA splicing via spliceosome' ($P = 0.0084$), suggesting it captures a more specific association within this pathway. Among the 153 unique genes across these two genesets, 30 reached nominal significance ($P < 0.05$), with top genes including *HSPA8*, *HABP4*, *NOVA2*, *HNRNPL* and *SNW1* (Supplementary Fig. 7a). We also performed a geneset analysis among the 51 ALS-linked genes curated by GCEP[14]. As expected, this showed that the 'Definitive' category was highly significant ($P < 1 \times 10^{-16}$) across allele frequency thresholds, whereas the 'Limited' category showed only modest enrichment ($P = 0.0015$), and no enrichment was seen among the other categories (Supplementary Fig. 7b,c).

We next examined whether carrying several variants among 'Definitive' ALS genes as classified by GCEP confers cumulative risk. We observed a clear dose–response relationship across low-frequency variants (MAF < 0.05): the OR increased progressively as people carried one (OR = 1.19, $P = 2.11 \times 10^{-15}$), two (OR = 1.35, $P = 8.43 \times 10^{-13}$), three (OR = 1.84, $P = 2.78 \times 10^{-8}$) or four (OR = 4.26, $P = 5.35 \times 10^{-5}$) qualifying variants (Extended Data Fig. 4c and Supplementary Data 9).

This relationship persisted when burden was assessed at the gene level, where several variants within the same gene were counted as a single event (Extended Data Fig. 4d and Supplementary Data 9). Analyses restricted to rarer variants were underpowered due to the low number of people carrying several variants (Supplementary Data 9). We did not observe a similar dose–response relationship when we tested for an association with age at onset and survival (Supplementary Fig. 8).

We next focused on co-occurrence among the specific risk variants identified in this study. Focusing on single variants in 'Definitive' GCEP genes, we found that 11.1% of cases carried one variant and 0.54% carried two, whereas the co-occurrence of three or more variants was not observed (Extended Data Table 3). When including variants in genes with 'Limited' evidence and new single variants identified in this study, the proportions increased to 14.5% for one, 1.1% for two and 0.0076% for three variants. The proportions increased further to 18.2%, 1.7% and 0.099%, respectively, when also including qualifying variants from the URV burden analyses. Finally, when *C9orf72* repeat expansion status was also considered (available for 66% of cases), these totals rose to 23.5%, 3.12% and 0.22%, respectively, totaling 26.9% of cases. The observed co-occurrence rates did not deviate from those expected under an additive model using permutation analyses ($P = 0.39$). When examining specific variant pairs, we observed numerous instances of cases carrying several variants (Extended Data Fig. 5 and Supplementary Fig. 9). For example, 20% of *C9orf72* repeat expansion

**Table 1 | New rare single variants achieving significance**

| Variant | Gene | Consequence | Discovery | | | | Replication | | | | Meta-analysis |
|---|---|---|---|---|---|---|---|---|---|---|---|
| | | | Case MAC (MAF) | Control MAC (MAF) | OR (95% CI) | P | Case MAC (MAF) | Control MAC (MAF) | OR (95% CI) | P | P |
| 9:133154147:C:A | GBGT1 | c.455G>T/p.R152L | 14 (5.34×10⁻⁴) | 3 (2.15×10⁻⁵) | 26.9 (9.19–104) | 1.68×10⁻¹⁰ | 1 (1.05×10⁻⁴) | 0 (0.00) | 44.7 (2.38–6,522) | 1.34×10⁻² | 1.99×10⁻¹¹ |
| 7:44206388:A:G | YKT6 | c.191A>G/p.Y64C | 48 (1.83×10⁻³) | 91 (6.52×10⁻⁴) | 2.84 (1.97–4.05) | 9.08×10⁻⁸ | 15 (1.57×10⁻³) | 164 (6.26×10⁻⁴) | 2.54 (1.37–4.37) | 4.15×10⁻³ | 1.50×10⁻⁹ |
| 1:223762207:A:G | CAPN2 | c.1588A>G/p.I530V | 12 (4.57×10⁻⁴) | 3 (2.15×10⁻⁵) | 25.3 (8.30–100) | 3.66×10⁻⁹ | 1 (1.05×10⁻⁴) | 3 (1.15×10⁻⁵) | 5.11 (0.434–38.4) | 1.69×10⁻¹ | 1.18×10⁻⁸ |
| 3:184057041:A:G | HTR3C | c.556A>G/p.T186A | 39 (1.48×10⁻³) | 70 (5.05×10⁻⁴) | 3.41 (2.27–5.04) | 1.87×10⁻⁸ | 6 (6.28×10⁻⁴) | 97 (3.71×10⁻⁴) | 2.18 (0.868–4.57) | 9.15×10⁻² | 1.71×10⁻⁸ |
| 12:122547457:T:A | KNTC1 | c.859T>A/p.W287R | 9 (3.43×10⁻⁴) | 2 (1.43×10⁻⁵) | 27.7 (7.80–145) | 1.07×10⁻⁷ | 1 (1.05×10⁻⁴) | 2 (7.64×10⁻⁶) | 12.2 (1.05–106) | 4.67×10⁻² | 2.96×10⁻⁸ |

Listed are new variants that reached significance in the exome-wide discovery analysis ($P < 1.83 \times 10^{-7}$). Test statistics are shown for the discovery phase ($n_{cases} = 13{,}138$; $n_{controls} = 69{,}775$), replication phase ($n_{cases} = 4{,}781$; $n_{controls} = 130{,}928$) and the combined meta-analysis (Stouffer's Z score method, weighted by effective sample size). Association statistics were estimated using Firth's logistic regression with profile penalized likelihood CIs. P values are two-tailed and presented uncorrected for multiple testing.

carriers harbored additional risk variants. Furthermore, some pairs, including *CFAP410* p.V58L × *NEK1* p.R261H, showed trends suggestive of a synergistic effect (Extended Data Fig. 4e). To formally test whether any of these pairs showed nonadditive effects, we performed pairwise co-occurrence and interaction analyses. No pairs reached significance after correction for multiple testing (Extended Data Fig. 5 and Supplementary Fig. 9). This was consistent with our power calculations (Supplementary Fig. 10), which showed that the study was underpowered to detect all but the largest deviations from additivity for specific variant pairs, and then only for pairs including at least one low-frequency variant (0.01 < MAF < 0.05).

### *ARPP21* p.P563L is associated with earlier disease onset and shorter disease duration

To assess the impact of genetic variants on disease progression, we analyzed survival and age at onset across candidate genes and variants (Extended Data Fig. 6 and Supplementary Data 10). Consistent with previous reports, *SOD1* p.A5V and p.D91A were associated significantly with a lower age at onset (p.A5V: B = −9.44, $P = 5.61 \times 10^{-4}$; p.D91A: B = −4.82, $P = 1.11 \times 10^{-5}$), with p.A5V linked to shorter survival and p.D91A to longer survival (p.A5V: hazard ratio (HR) = 13.0, $P = 1.19 \times 10^{-8}$; p.D91A: HR = 0.453, $P = 1.48 \times 10^{-7}$). Similarly, *FUS* p.R521C and p.P525L were associated with earlier onset (p.R521C: B = −16.2, $P = 1.90 \times 10^{-4}$; p.525L: B = −39.1, $P = 1.53 \times 10^{-10}$), with p.P525L specifically associated with shorter survival (HR = 41.75, $P = 1.41 \times 10^{-10}$). Notably, *ARPP21* p.P563L was associated with a significantly lower age at onset (B = −12.7, $P = 5.44 \times 10^{-4}$) and shorter survival (HR = 5.96, Δsurvival time = −19.5 months, $P = 2.54 \times 10^{-6}$), showing effect sizes comparable to *SOD1* p.A5V (Extended Data Fig. 6a). Among URVs, *SOD1* was associated with longer survival (HR = 0.45, $P = 0.0022$), whereas no significant associations were observed for other genes (Extended Data Fig. 6b).

### Replication confirms *YKT6* and supports *HTR3C*, *GBGT1* and *KNTC1* as ALS risk genes

For replication, we generated a cohort comprising 4,781 individuals with ALS and 130,928 controls after applying stringent quality control criteria identical to those used in the discovery set (Supplementary Figs. 11 and 12). Power analyses based on the (winner's curse adjusted) effect sizes observed in the discovery dataset indicated that this provides between 32% and 91% statistical power for replication across candidate variants and genes (Supplementary Fig. 12).

Of the five new single variants identified in the discovery phase, all showed a consistent direction of effect in the replication cohort (Table 1; $\lambda_{1,000} = 0.965$). Moreover, all five reached exome-wide significance in a meta-analysis of the combined discovery and replication data, with all but *CAPN2* p.I530V showing greater significance compared to the discovery phase alone (Table 1). Furthermore, *YKT6* p.Y64C achieved replication-wide significance ($P < 0.0063$), correcting for the eight new associations from the discovery phase (five single variants and three URV genes). Among the three candidate URV genes, a consistent direction of effect was seen only for *KIF4A* (OR = 2.46, $P = 0.26$), and none reached replication-wide significance (Table 2; $\lambda_{1,000} = 1.046$).

### Establishing independent evidence for *ARPP21*, *DNAJC7* and *CFAP410*

Next, for the genes that were significant in our discovery analysis that are currently classified by GCEP with 'Limited' evidence (*ARPP21*, *CFAP410* and *DNAJC7*), we aimed to confirm the independence of our findings.

For *ARPP21*, we identified two rare variants: p.P747L, which has not previously been reported in the scientific literature, and p.P563L, previously reported in UK and Spanish families as candidate variants[18,19]. To confirm independence for p.P563L, we excluded four potentially overlapping UK carriers (no Spanish carriers were identified). The association remained (OR = 28.3, $P = 3.47 \times 10^{-7}$; Extended Data Fig. 7a)

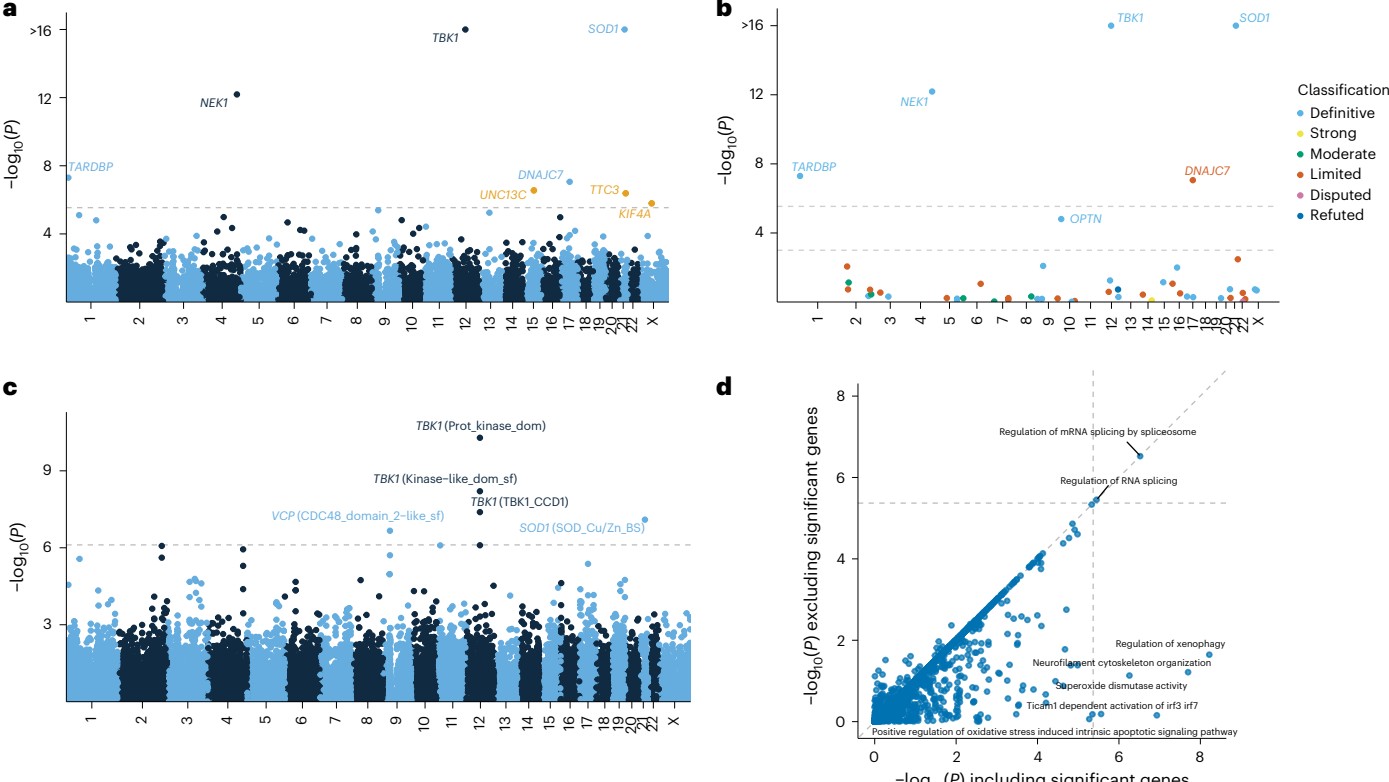

**Fig. 2 | URV burden analyses. a**, *y* axis: exome-wide gene-based URV associations (−log₁₀(*P*)); *x* axis: genomic coordinates (GRCh38). Dashed line: exome-wide significance threshold (*P* < 2.9 × 10⁻⁶). New risk genes are highlighted in orange. **b**, URV burden analyses among ALS-linked genes curated by the ALS GCEP. *y* axis: gene-based URV associations (−log₁₀(*P*)); *x* axis: genomic coordinates (GRCh38). Lower dashed line: significance threshold across ALS-linked genes (*P* < 1 × 10⁻³); upper dashed line: exome-wide significance threshold as presented in **a**. **c**, Domain-based URV analyses. *y* axis: domain associations (−log₁₀(*P*)); *x* axis:

genomic coordinates (GRCh38). Dashed line: exome-wide significance threshold (*P* < 7.68 × 10⁻⁷). **d**, Association *P* values for URV geneset burden analyses excluding exome-wide significant genes (*y* axis) versus including exome-wide significant genes (*x* axis). The dashed lines indicate the multiple testing threshold (*P* < 4.25 × 10⁻⁶). Association statistics were estimated using Firth's logistic regression with profile penalized likelihood CIs. *P* values are from the ACAT omnibus test combining the four variant filtering strategies (Methods) and are two-tailed and uncorrected for multiple testing.

and was further supported by our replication dataset, which had no potential overlap with previous studies (OR = 16.5, *P* = 3.29 × 10⁻³). A meta-analysis of these two independent datasets yielded a highly significant association (*P* = 4.31 × 10⁻⁹), confirming a strong, independent signal. We also validated the reported effects of age of onset and progression[19] in our nonoverlapping cohort (Extended Data Fig. 6). Finally, *ARPP21* carriers were observed across several cohorts beyond those from the UK and Spain, significantly expanding its known population distribution (Supplementary Fig. 5).

*CFAP410* p.V58L was previously identified in two common variant genome-wide association studies (GWAS) (MAF = 0.013)[4,20]. To confirm independence, we excluded 8,372 cases and 4,159 controls that were duplicated or had second-degree or closer genetic relatedness to the original GWAS cohorts. The association remained highly significant after this exclusion (*P*_meta = 1.34 × 10⁻¹⁴), with consistent ORs in both discovery (OR = 1.81, *P* = 1.32 × 10⁻¹⁰) and replication (OR = 1.61, *P* = 1.09 × 10⁻⁵) cohorts (Extended Data Fig. 7b).

For *DNAJC7*, implicated previously in a case–control study of ALS[21], we re-evaluated the association after excluding overlapping cohorts (excluding 5,722 cases and 9,849 controls). In this reduced discovery dataset, there remained a robust association with a consistent odds ratio (*n*_cases = 7,606, *n*_controls = 59,926; OR = 2.56, *P* = 1.36 × 10⁻⁴; Extended Data Fig. 7c). This was further supported by our replication cohort, which had minimal overlap (190 cases) with the previous study (OR = 2.41, *P* = 2.82 × 10⁻³; Extended Data Fig. 7c). Meta-analysis across these two datasets confirmed a strong, independent signal (*P* = 2.96 × 10⁻⁶).

## Discussion

This study represents one of the largest rare variant analysis of ALS until now, capturing nearly all of its known rare variant architecture. We expand upon known genetic factors contributing to ALS, demonstrating a substantially higher yield than common variant GWAS of comparable size[4,20].

Our findings also provide a broader view of the genetic architecture of ALS. The variants we identified were mostly missense and spanned a spectrum of effect sizes, ranging from low-frequency variants with moderate effect sizes and URVs conferring large effects. At the far end of this spectrum, the latter category includes variants such as *SOD1* p.A5V, which was absent from ~200,000 controls and conferred large effects (OR > 200), alongside other high impact variants such as those in *ARPP21* (ORs > 40). Moreover, our data support an additive, oligogenic model[22,23] wherein several rare variants cumulatively increase ALS risk without strong evidence of interaction. Although statistical power was limited, pairwise analyses of the best-powered variant combinations revealed no significant interactions, consistent with an additive model. Fully characterizing this oligogenic architecture will require studies extending beyond established ALS genes and, ultimately, beyond the exome.

Among the identified rare variants, *YKT6* p.Y64C stood out due to its highly significant and consistent associations in both the discovery and replication cohorts. It was associated with a moderate increase in risk, with an OR comparable to that of established ALS variants *SOD1* p.D91A and *NEK1* p.R261H (Fig. 1c). *YKT6* encodes a highly conserved SNARE protein that plays a key role in vesicular transport pathways,

**Table 2 | New genes achieving exome-wide significance in URV burden analyses**

| Gene | Filtering strategy | Discovery | | | | Replication | | | | Meta-analysis |
| --- | --- | --- | --- | --- | --- | --- | --- | --- | --- | --- |
| | | No. of case carriers (frequency) | No. of control carriers (frequency) | OR (95% CI) | P | No. of case carriers (frequency) | No. of control carriers (frequency) | OR (95% CI) | P | P |
| KIF4A | Moderate-impact/ singletons only | 22 (1.68×10⁻³) | 27 (3.90×10⁻⁴) | 4.69 (2.61–8.33) | 1.62×10⁻⁶ | 2 (4.21×10⁻⁴) | 52 (4.01×10⁻⁴) | 2.74 (0.538–8.59) | 8.75×10⁻¹ | 2.44×10⁻⁶ |
| TTC3 | Moderate-impact/ ultrarare | 152 (1.17×10⁻²) | 380 (5.53×10⁻³) | 1.73 (1.42–2.10) | 4.16×10⁻⁷ | 44 (9.21×10⁻³) | 938 (7.28×10⁻³) | 0.812 (0.568–1.13) | 3.90×10⁻² | 5.62×10⁻⁴ |
| UNC13C | Moderate-impact/ ultrarare | 200 (1.56×10⁻²) | 585 (8.61×10⁻³) | 1.59 (1.35–1.87) | 2.80×10⁻⁷ | 58 (1.21×10⁻²) | 1,075 (8.30×10⁻³) | 0.750 (0.540–1.02) | 1.90×10⁻¹ | 8.62×10⁻⁴ |

Listed are new genes that reached significance (P < 2.89×10⁻⁶) in the URV burden analyses. Test statistics are shown for the discovery phase ($n_{cases}$ = 13,138; $n_{controls}$ = 69,775), replication phase ($n_{cases}$ = 4,781; $n_{controls}$ = 130,928) and the combined meta-analysis (Stouffer's Z-score method, weighted by effective sample size). Carrier frequencies, ORs and CIs estimated using Firth's logistic regression with profile penalized likelihood confidence intervals are presented for the most significant of the four variant filtering strategies. P values are two-tailed, uncorrected for multiple testing, and estimated using the ACAT omnibus test combining the four variant filtering strategies.

also implicated in GWAS[20], and is critical for autophagosome-lysosome fusion[24,25]. The p.Y64C variant has been linked to a neurodevelopmental disorder in the homozygous state[26], whereas we found it to be associated with ALS in the heterozygous state. The variant was shown to cause partial loss-of-function and impaired autophagy in *Drosophila*[26], in line with it being consistently predicted as damaging by all in silico predictors we tested. YKT6 has also been implicated in the secretion of the MSP domain of the ALS-linked protein VAPB, suggesting that it may also play a role in extracellular signaling[27,28]. The identification of *YKT6* thus highlights the central role of disrupted vesicle fusion and trafficking in ALS and specifically implicates downstream consequences including impaired autophagy and exocytosis. Beyond *YKT6*, three high-effect (OR > 20) missense variants in *KNTC1*, *HTR3C* and *GBGT1* also represent strong candidates displaying consistent directions of effect across discovery and replication analyses, with meta-analysis achieving greater statistical significance than the discovery analysis alone. These genes implicate GTPase signaling, serotonergic function and glycosphingolipid metabolism respectively, all processes previous linked to ALS pathology[29–32].

In addition to these new findings, a key contribution of our study is providing robust, independent evidence for several genes with limited previous evidence. In *ARPP21*, we identified two high-effect variants (p.P563L, p.P747L), with ORs comparable to those of highly penetrant variants such as *FUS* p.R521C and *TARDBP* p.N352S (Fig. 1c). Of these, p.P563L has been reported previously in two family studies[18,19]. Our study now firmly establishes not only the association of this variant with ALS, but also its previously reported effects on age of onset and survival. Moreover, this variant had been reported only in UK and Spanish families, whereas our study establishes its relevance in a broader population, identifying carriers across Dutch, US, Italian and Israeli cohorts. The second variant, p.P747L, has not previously been reported in the scientific literature. ARPP21, like TDP-43 and FUS, is an RNA-binding protein that localizes to stress granules under stress[33]. Our observed enrichment of URVs in splicing-related genes adds support to the relevance of this commonality, and suggests further insights remain to be discovered concerning the full depth of RNA processing dysfunction in ALS pathogenesis. For *DNAJC7*, which encodes a heat-shock protein implicated previously in an exome-wide burden analysis[21], our study offers independent validation. Although the previous signal was driven by protein-truncating variants, our URV analysis identified a robust association that also included INDELs and missense variants. Crucially, this association was replicated across cohorts and remained after removal of cohorts overlapping with the previous study. Finally, in *CFAP410* (also known as *C21orf2*), the low-frequency (MAF = 0.013) missense variant p.V58L was identified previously in two common variant GWASs[4,20] and has been linked to primary cilia dysfunction[34]. We show that this is a robust and independent finding, as the association remained highly significant even after we excluded all participants who were duplicated or genetically related to the original GWAS cohorts.

Finally, despite inconclusive evidence from the replication analysis, *UNC13C*, *KIF4A* and *CAPN2* remain candidates of interest for further study. *UNC13C* and *KIF4A* are paralogs of ALS genes *UNC13A* and *KIF5A*, respectively[35–38], and are similarly involved in synaptic vesicle release and axonal transport (Supplementary Table 2 and Supplementary Data 11). *CAPN2* is of interest due to previous literature supporting its role in ALS pathology and is being evaluated as a therapeutic target for antisense oligonucleotide (ASO) therapies[39–41]. Further investigation of these genes in independent datasets is warranted to fully elucidate their potential role in ALS.

Our findings have clear translational potential. ASO-based therapies such as Tofersen (*SOD1*) and Jacifusen (*FUS*)[42] demonstrate the feasibility of gene-targeted treatment, and ongoing individualized approaches (for example, Silence ALS, n-Lorem) extend this to URVs. Our study increases the proportion of cases with an identifiable genetic risk factor from 11.6% ('Definitive' genes) to 15.6% with our validated

and new single-variants, a figure that rises to 22.9% when *C9orf72* repeat expansions are included. Although not all identified genes will be viable ASO targets, as evidenced by setbacks in trials targeting *ATXN2* and *C9orf72* (ref. 43), genes harboring variants with high ORs such as *ARPP21* represent prime candidates to prioritize for future ASO-based studies.

Our study has limitations. First, we did not perform functional validation for the identified variants; therefore, the precise molecular mechanisms (for example, loss-of-function, gain-of-function or a combination thereof) remain to be determined. Second, by design, the exome analyses in this study precluded the investigation of noncoding variation. However, it is important to note that about three-quarters of the cases included in the discovery cohort and all replication cohorts underwent whole-genome sequencing from which exomes were derived in silico. In addition, the increasing availability of large-scale WGS data, particularly in the UK Biobank and All of Us, provides ample controls, meaning that much of what is needed to build large-scale WGS datasets for future analyses is already available[44,45]. Third, our analysis is restricted to germline variants and therefore does not address the potential role of somatic mutations, which accumulate in the central nervous system with aging and could therefore explain the late-onset of the disease[46]. Fourth, we acknowledge the importance of expanding beyond the predominantly European ancestry of participants included in this study. Finally, although our study captures most of the known ALS genetic architecture, a few genes remain undetected. These are either genes associated primarily with repeat expansions (*ATXN2* and *C9orf72*) or genes harboring exceedingly rare variants (*VAPB* and *PFN1*).

To conclude, the assembly of the largest exome sequencing dataset for ALS to date, coupled with robust harmonization and replication, enabled the discovery of rare variant contributions to ALS. We show that rare variant analyses yield particularly high returns in ALS compared to common variant GWAS. The identification of several new genes, alongside the confirmation of genes with previous limited evidence, collectively provides a compelling set of potential new targets for translational ALS research.

## Online content

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

Paul J. Hop [ORCID][1], Maarten Kooyman [ORCID][1], Brendan J. Kenna [ORCID][1], Ramona A. J. Zwamborn[1], Kristel R. van Eijk[1], Yan Wang[2], Charlotte H. van Dijk [ORCID][2], Erwin Bekema[1], Wouter van Rheenen [ORCID][1], Paul Beele [ORCID][1], Joke J. F. A. van Vugt [ORCID][1], Project MinE ALS sequencing Consortium*, NYGC ALS Consortium*, FALS sequencing Consortium*, GTAC Consortium*, Ahmad Al Khleifat [ORCID][3], Alfredo Iacoangeli [ORCID][4], Johnathan Cooper-Knock [ORCID][5,6], Bradley N. Smith [ORCID][3], Simon Topp [ORCID][4], Anneke J. van der Kooi[7], Vera Fominykh [ORCID][8], Vivian Drory[9], Yossef Lerner[10], Yehuda Shovman [ORCID][10], Dominic B. Rowe[11], Kelly L. Williams [ORCID][11], Russell L. McLaughlin [ORCID][12], Jessica Hurt[13], Yunfeng Huang [ORCID][13], Chia-Yen Chen [ORCID][13], Ellen Tsai [ORCID][13], Heiko Runz [ORCID][13], Eleonora Aronica [ORCID][14], Ewout J. N. Groen [ORCID][1], Michael A. van Es [ORCID][1], R. Jeroen Pasterkamp [ORCID][2], Sali M. K. Farhan[15,16,17], Fleur C. Garton[18], Allan F. McRae[18], Pamela A. McCombe[18], Robert D. Henderson[18], Dongsheng Fan[19,20], Lenka Šlachtová [ORCID][21], Helle Høyer [ORCID][22,23], Agnes L. Nishimura[24,25], Ruben J. Cauchi [ORCID][26], Lev Brylev[27], Boris Rogelj [ORCID][28,29], Blaž Koritnik [ORCID][30], Janez Zidar[30], Teresa Salas[31], Jesus S. Mora Pardina[32], Marc Gotkine [ORCID][10], Monica Povedano[33], Philippe Corcia[34], Patrick Vourc'h[35], Philippe Couratier[36], Markus Weber [ORCID][37], Matthew C. Kiernan[38], Roger Pamphlett[39], Ian P. Blair[11], Mamede de Carvalho [ORCID][40], Nazli A. Başak [ORCID][41], Caroline Ingre[42,43], Peter M. Andersen [ORCID][44], Lorne Zinman[45,46], Ekaterina Rogaeva [ORCID][47], Ian R. MacKenzie [ORCID][48], Nicolas Dupre[49,50], Guy A. Rouleau [ORCID][51], Bryan J. Traynor[52,53], Nicola Ticozzi [ORCID][54,55], Adriano Chiò[56], Vincenzo Silani [ORCID][54,55], Orla Hardiman[57], Hemali Phatnani[58,59,60], Matthew B. Harms [ORCID][58,59,61], Clifton L. Dalgard [ORCID][62], Jonathan D. Glass [ORCID][63], John E. Landers [ORCID][64], Philip Van Damme [ORCID][65], Karen E. Morrison [ORCID][66], Pamela J. Shaw [ORCID][5,6], Chris E. Shaw[3], Ammar Al-Chalabi [ORCID][3], Leonard H. van den Berg [ORCID][1], Kevin P. Kenna [ORCID][2,67] ✉ & Jan H. Veldink [ORCID][1,67] ✉

¹Department of Neurology, UMC Utrecht Brain Center, University Medical Center Utrecht, Utrecht University, Utrecht, The Netherlands. ²Department of Translational Neuroscience, UMC Utrecht Brain Center, University Medical Center Utrecht, Utrecht, The Netherlands. ³Department of Basic and Clinical Neuroscience, Maurice Wohl Clinical Neuroscience Institute, King's College London, London, UK. ⁴Department of Biostatistics and Health Informatics, Institute of Psychiatry, Psychology and Neuroscience, King's College London, London, UK. ⁵Sheffield Institute for Translational Neuroscience (SITraN), University of Sheffield, Sheffield, UK. ⁶NIHR Sheffield Biomedical Research Centre, Sheffield, UK. ⁷Department of Neurology, Amsterdam University Medical Center, Amsterdam Neuroscience, Amsterdam, The Netherlands. ⁸Centre for Precision Psychiatry, Institute of Clinical Medicine, University of Oslo, Oslo, Norway. ⁹Department of Neurology, Neuromuscular Diseases Unit, Tel Aviv Sourasky Medical Center, Tel Aviv, Israel. ¹⁰Department of Neurology, Hadassah Medical Organization and Faculty of Medicine, Hebrew University of Jerusalem, Jerusalem, Israel. ¹¹Motor Neuron Disease Research

Centre, Macquarie Medical School, Faculty of Medicine, Health and Human Sciences, Macquarie University, Sydney, New South Wales, Australia. [12]Complex Trait Genomics Laboratory, Smurfit Institute of Genetics, Trinity College Dublin, Dublin, Republic of Ireland. [13]Research, Biogen, Cambridge, MA, USA. [14]Department of (Neuro)Pathology, Amsterdam UMC, University of Amsterdam, Amsterdam, The Netherlands. [15]Department of Neurology and Neurosurgery, McGill University, Montreal, Quebec, Canada. [16]Montreal Neurological Institute-Hospital, McGill University, Montreal, Quebec, Canada. [17]Department of Genetics, McGill University, Montreal, Quebec, Canada. [18]Centre for Clinical Research, University of Queensland, Brisbane, Queensland, Australia. [19]Department of Neurology, Peking University Third Hospital, Beijing, China. [20]Beijing Municipal Key Laboratory of Biomarker and Translational Research in Neurodegenerative Diseases, Beijing, China. [21]Institute of Biology and Medical Genetics, First Faculty of Medicine, Charles University in Prague, Prague, Czech Republic. [22]Department of Medical Genetics, Telemark Hospital Trust, Skien, Norway. [23]Faculty of Medicine,  Institute of Clinical Medicine, University of Oslo, Oslo, Norway. [24]Centre for Neuroscience, Surgery and Trauma, Blizard Institute, Barts and The London School of Medicine and Dentistry, Queen Mary University of London, London, UK. [25]Paulo Gontijo Institute, São Paulo, Brazil. [26]Centre for Molecular Medicine and Biobanking and Department of Physiology and Biochemistry, Faculty of Medicine and Surgery, University of Malta, Msida, Malta. [27]New York University Abu Dhabi, Abu Dhabi, United Arab Emirates. [28]Department of Biotechnology, Jozef Stefan Institute, Ljubljana, Slovenia. [29]Faculty of Chemistry and Chemical Technology, University of Ljubljana, Ljubljana, Slovenia. [30]Ljubljana ALS Centre, University Medical Centre Ljubljana, Institute of Clinical Neurophysiology, Ljubljana, Slovenia. [31]Department of Neurology, Hospital La Paz-Carlos III, Madrid, Spain. [32]Affiliation ALS Unit, Hospital Universitario San Rafael, Madrid, Spain. [33]La Unitat Funcional de Motoneurona, Cap de Secció de Neurofisiologia, Servei de Neurologia, Hospital Universitario de Bellvitge-IDIBELL, L'Hospitalet de Llobregat, Spain. [34]Centre SLA, CHRU de Tours, UMR 1253, iBrain, Université de Tours, Inserm, Tours, France. [35]Service de Biochimie et Biologie moléculaire, CHU de Tours, Tours, France. [36]Centre SLA CHU Dupuytren, Limoges, France. [37]Neuromuscular Diseases Unit/ALS Clinic, Kantonsspital St. Gallen, St. Gallen, Switzerland. [38]Neuroscience Research Australia, University of New South Wales, Sydney, New South Wales, Australia. [39]University of Sydney, Royal Prince Alfred Hospital, Sydney, New South Wales, Australia. [40]Instituto de Fisiologia, Instituto de Medicina Molecular, Faculdade de Medicina, Universidade de Lisboa, Lisbon, Portugal. [41]School of Medicine, Molecular Biology and Genetics- KUTTAM, Koç University, Suna and Inan Kıraç Foundation, Istanbul, Turkey. [42]Department of Clinical Neuroscience, Karolinska Institutet, Stockholm, Sweden. [43]Department of Neurology, Karolinska University Hospital, Stockholm, Sweden. [44]Department of Clinical Science, Neurosciences, Umeå University, Umeå, Sweden. [45]Sunnybrook Health Sciences Centre, Toronto, Ontario, Canada. [46]Division of Neurology, University of Toronto, Toronto, Ontario, Canada. [47]Tanz Centre for Research in Neurodegenerative Diseases, University of Toronto, Toronto, Ontario, Canada. [48]Department of Pathology, University of British Columbia, Vancouver, British Columbia, Canada. [49]Neuroscience axis of CHU de Québec - Université Laval, Quebec City, Quebec, Canada. [50]Department of Medicine, Faculty Medicine, Laval University, Quebec City, Quebec, Canada. [51]Montreal Neurological Institute and Hospital, McGill University, Montreal, Quebec, Canada. [52]Neuromuscular Diseases Research Section, Laboratory of Neurogenetics, National Institute on Aging, NIH, Porter Neuroscience Research Center, Bethesda, MD, USA. [53]Department of Neurology, Johns Hopkins University Medical Center, Baltimore, MD, USA. [54]Department of Neurology and Laboratory of Neuroscience, Istituto Auxologico Italiano IRCCS, Milan, Italy. [55]Department of Pathophysiology and Transplantation, 'Dino Ferrari' Center, Università degli Studi di Milano, Milan, Italy. [56]'Rita Levi Montalcini' Department of Neuroscience, ALS Centre, University of Torino, Turin, Italy. [57]Academic Unit of Neurology, Trinity College Dublin, Trinity Biomedical Sciences Institute, Dublin, Republic of Ireland. [58]Department of Neurology, Columbia University Irving Medical Center, New York, NY, USA. [59]Center for Motor Neuron Biology and Disease, Columbia University Irving Medical Center, New York, NY, USA. [60]New York Genome Center, New York, NY, USA. [61]Institute for Genomic Medicine, Columbia University Irving Medical Center, New York, NY, USA. [62]The American Genome Center, Uniformed Services University—'America's Medical School', Bethesda, MD, USA. [63]Department of Neurology, Emory University School of Medicine, Atlanta, GA, USA. [64]Department of Neurology, UMass Chan Medical School, Worcester, MA, USA. [65]Department of Neurosciences, and Department of Neurology,  KU Leuven—University of Leuven, University Hospitals Leuven and Leuven Brain Institute (LBI), Leuven, Belgium. [66]School of Medicine, Dentistry and Biomedical Sciences, Queen's University Belfast, Belfast, UK. [67]These authors jointly supervised this work: Kevin P. Kenna, Jan H. Veldink. *Lists of authors and their affiliations appear at the end of the paper.
✉e-mail: K.P.Kenna@umcutrecht.nl; J.H.Veldink@umcutrecht.nl

## Project MinE ALS sequencing Consortium

Philip Van Damme[65], Philippe Corcia[34], Philippe Couratier[36], Patrick Vourc'h[35], Orla Hardiman[57], Russell L. McLaughlin[12], Marc Gotkine[10], Vivian Drory[9], Nicola Ticozzi[54,55], Vincenzo Silani[54,55], Jan H. Veldink[1,67], Leonard H. van den Berg[1], Mamede de Carvalho[40], Jesus S. Mora Pardina[32], Monica Povedano[33], Peter M. Andersen[44], Markus Weber[37], Nazli A. Başak[41], Ammar Al-Chalabi[3], Chris E. Shaw[3], Pamela J. Shaw[5,6], Karen E. Morrison[66], John E. Landers[64], Jonathan D. Glass[63] & Clifton L. Dalgard[62]

## NYGC ALS Consortium

Hemali Phatnani[58,59,60], Matthew B. Harms[58,59,61], Eleonora Aronica[14], Marc Gotkine[10], Vivian Drory[9], Yossef Lerner[10] & Bryan J. Traynor[52,53]

## FALS sequencing Consortium

Kevin P. Kenna[2,67], Bradley N. Smith[3], Simon Topp[4], Ammar Al-Chalabi[3], Leonard H. van den Berg[1], Jan H. Veldink[1,67], Vincenzo Silani[54,55], Nicola Ticozzi[54,55], John E. Landers[64], Chris E. Shaw[3], Jonathan D. Glass[63] & Guy A. Rouleau[51]

## GTAC Consortium

Matthew B. Harms[58,59,61]

## Methods

### Cohorts

This study was approved by the institutional review boards of all participating centers, written informed consent for research was obtained from each participant and the study was approved by the Medical Ethical Testing Committee NedMec and the Biobanks Testing Committee of UMC Utrecht. Cases were included in this study irrespective of their carrier status for variants in known ALS genes.

**Discovery cohort.** The discovery cohort included 15,862 participants with ALS and 78,683 controls, totaling 94,545 individuals, of which 21,102 were subjected to WGS and 73,443 to WXS. Case cohorts included the Project MinE ALS sequencing consortium (7,614 cases; 2,605 controls)[47], the NYGC ALS Consortium (2,650 cases; 342 controls), the ALS Sequencing Consortium (2,851 cases)[5], two cohorts from the FALS consortium (1,277 cases; phs001585), the National Institutes of Health (NIH) Exome Sequencing of FALS Project (194 cases; phs000101), two Australian cohorts described in ref. 48 (125 cases, 18 controls) and ref. 49 (568 cases), and a Chinese motor neurone disease (MND) cohort[50] (583 cases, 182 controls). All cases were diagnosed with definite, probable or probable laboratory-supported ALS according to the revised El Escorial Criteria[51]. Control cohorts included 7,323 samples from the National Heart, Lung, and Blood Institute (NHLBI) TOPMed research program[52], 49,981 samples from the UK Biobank[44] and 18,232 samples across seven cohorts from dbGAP[53].

**Replication cohort.** The replication cohort included 5,404 people with ALS and 133,823 controls, totaling 139,227 participants, all of whom were subjected to WGS. Cohorts include the Project MinE ALS sequencing consortium (1,510 cases; 169 controls), the NYGC ALS consortium (1,257 cases; 69 controls), ALS compute (1,870 cases; 1,820 controls; phs003184) and the UK Biobank (767 cases; 131,765 controls). During sample quality control, people who were duplicates or related up to the second degree to any participant in the discovery cohort were excluded.

### Processing of sequencing data

Processing and annotation of sequencing data was performed as described previously[54]. All raw sequencing data were aligned to the GRCh38 reference genome using BWA-mem[55] according to the functional equivalence pipeline described by Regier et al.[11] (implementation can be found at https://github.com/maarten-k/realignment). Joint genotyping was performed using a uniform pipeline according to the GATK best practices (v.4.2.6.1)[10]. Genotype calls with a quality score < 20 were set to missing, variant calls supported by uninformative reads were excluded and multiallelic variants were split into biallelic variants. Male genotypes in nonpseudoautosomal regions on chromosome X were coded as 0 or 1 (according to 0 or 1 allele copies).

### Variant annotation

Variants were annotated using snpEff[56], dbscSNV[57] and Ensembl Release v.105 gene models[58]. Variants were classified as high-impact when predicted by snpEff to have a high impact (including nonsense mutations, splice acceptor/donors and frameshift mutations) or predicted as potentially splice-altering by dbscSNV ('ada' or 'rf' score > 0.7). Variants were classified as having moderate impact when predicted as such by snpEff (including missense mutations, in-frame deletions and UTR truncations). For each gene, the impact of a variant was determined by its most severe consequence across protein-coding transcripts.

### Sample quality control

Ancestry was estimated by projecting all samples on a reference ancestry space comprising samples from the 1000 Genomes project using the LASER software (v.2.04)[59]. We retained participants of predominantly European ancestry. We then excluded samples with low genotype call-rate (<0.9), discordant sex or deviating heterozygosity (inbreeding $F < -0.1$ or $F > 0.1$). These metrics were calculated in a set of autosomal variants meeting the following criteria: call-rate > 0.9 in each supercohort (discovery: WGS, WXS$_{UKB}$, WXS$_{other}$; replication: Project MinE, ALS compute, NYGC, UK Biobank), MAF > 0.01 and, for sex inference, heterozygosity, relatedness and PCA variants were also filtered based on Hardy–Weinberg equilibrium (HWE) ($P < 0.0001$; for nonpseudoautosomal regions on chromosome X, these were calculated among female participants only) and pruned if in linkage disequilibrium (LD) ($r^2 < 0.5$, window size = 50, step = 5; furthermore, high LD regions were excluded before PCA[60]). We then excluded samples based on a high exome-wide number of SNVs, INDELs, singletons, high INDEL/SNV ratio or deviating Ti/Tv ratio (thresholds listed in Supplementary Fig. 2). Sample duplicates and relatives up to and including the second degree were identified using KING software[61]. An unrelated sample set was generated by first excluding samples with five or more relations, followed by iteratively excluding participants with the highest number of relations, resolving ties by prioritizing (in order) ALS over controls and WGS over WXS samples. Furthermore, in the replication cohort, samples that were duplicated or related up to the second degree to any sample in the discovery cohort were excluded. PCA was performed on the unrelated sample set using fastPCA as implemented in plink2 (ref. 62). In the discovery cohort, a distinct cluster was identified on the fourth and fifth PC consisting of an Amish population, which was excluded as the cluster contained only controls (Supplementary Fig. 2f).

### Variant quality control

First, GATK variant quality score recalibration was applied to all variants using the training data and annotations as recommended by the GATK best practices[10]. Variants were excluded if they did not pass variant quality score recalibration, their genotyping rate was <0.9 in any of the supercohorts (discovery: WGS, WXS$_{UKB}$, WXS$_{other}$; replication: Project MinE, ALS compute, NYGC, UK Biobank) or if they did not pass the HWE test in controls ($P < 0.0001$). We then also excluded variants with subpar quality scores and variants located in regions showing signs of batch effects. Potential batch effects were identified by testing whether variant allele counts were associated with cohort membership within control subjects. Firth's logistic regression with profile penalized likelihood CIs was used to perform these control–control analyses, adjusting for sex and four PCs[12]. This procedure was repeated for each cohort (that is, 1 = subject in respective cohort, 0 otherwise). In total, 16 cohorts were tested (including all WGS controls versus all WXS controls; cohorts with <100 controls were merged into one cohort) in the discovery cohort and four cohorts were tested in the replication cohort. The minimum $P$ value across these analyses was used as a metric to identify variants associating with probable batch effects. The stringency of various standard variant quality control filters was then increased to eliminate variants exhibiting batch associated calling bias while maintaining maximal sensitivity for unbiased variant calls (Supplementary Fig. 4). Identical thresholds were used for SNVs and INDELs and we also excluded long insertions and deletions (>50 base pairs) and variants coding the reference allele in spanning deletions.

### Single-variant analyses

Single-variant analyses were performed for all high and moderate impact variants with MAF < 0.05 and at least MAF > $5 \times 10^{-5}$ (272,925 variants). For each variant, we tested for an association between ALS status and MAC using Firth's logistic regression with profile penalized likelihood confidence intervals, which properly controls for type I error when testing rare variants in an unbalanced case–control setting[12,63–65]. We adjusted for sex, ten PCs and total number of rare synonymous variants in each participant. All tests were two-sided, and the Bonferroni correction was used to correct for multiple testing.

Candidate single-variant associations were screened for additional technical biases and excluded if (1) variant concordance <0.9 among

678 between-cohort duplicates included in the unfiltered dataset; (2) showed batch effects among case cohorts ($P_{\text{case-case}} < P_{\text{case-control}}$) based on the same procedure as used in the control–control analyses, where we tested for an association between cohort membership and MAC of the respective variant (that is, 1 = subject in respective cohort, 0 otherwise); (3) the minor allele was supported by one read in >25% of carriers; (4) significant heterogeneity ($P_{\text{het}} < 0.001$) between this study and a recent ALS common variant GWAS by van Rheenen et al.[20], for variants that overlap between both studies.

A targeted analysis was conducted on variants within 51 ALS-linked genes curated by the ALS GCEP (accessed December 2024)[14]. Certain exons of known ALS genes had lower call rate within subcohorts of the dataset; to provide a more complete investigation of known ALS genes, we therefore did not apply the per-supercohort call-rate filter for this analysis.

### URV burden analyses

URV burden analyses were performed using four filtering strategies based on two criteria: (1) variant frequency—either all URVs or singleton-only variants; (2) variant impact—either only high-impact variants (nonsense, splice acceptor/donor and frameshift mutations) or both high- and moderate-impact variants (missense mutations, in-frame deletions and UTR truncations). Burden analyses were performed by testing for an association between ALS status and the aggregate effect of minor alleles observed per sample per functional unit using Firth's logistic regression with profile penalized likelihood confidence intervals[12]. Sex, ten PCs and the total number of qualifying synonymous variants in each participant were included as covariates. Tests were retained if there were at least ten carriers across the functional unit tested. Test-statistics across the four filtering strategies were combined using the Cauchy method (ACAT), which is designed to combine results from several statistical tests[15]. Candidate associations were screened for potential technical biases by assessing biases among case cohorts using the same procedure as used in the control–control analyses. Genes where $P_{\text{case-case}} < P_{\text{case-control}}$ were flagged as potentially driven by technical variation.

**Genes.** Genes were defined using Ensembl gene models (release v.105), including only protein-coding genes without annotation errors.

**Domains.** Protein coordinates for Interpro domains, coiled coils, transmembrane helices, low complexity regions and cleavage sites were retrieved from Ensembl v.105 (http://dec2021.archive.ensembl.org/biomart/martview/)[58]. For each transcript, variants were annotated to domains by remapping both the domain coordinates and variant positions to coding sequence (CDS) relative coordinates using the *mapToCDS* method in RVAT[13]. Variants up to 12 base pairs from the CDS border (introns and UTRs) were mapped to the respective border. Domains that spanned more than 90% of the width of the transcript were excluded.

**Genesets.** To identify genesets or pathways associated with ALS, we performed geneset burden analysis on 13,347 GO, KEGG and Reactome genesets from the Molecular Signatures Database (MSigDB v.7.5)[17]. Genesets including fewer than 5 or more than 1,000 genes were excluded, resulting in a total of 11,777 tested genesets.

### Variant co-occurrence analyses

We tested for a cumulative effect of carrying several risk variants among moderate- and high-impact variants in genes classified as 'Definitive' according to GCEP[14]. Participants were grouped into categories based on the number of variants carried: 0 (reference group), 1, 2…*n* variants. Both heterozygous and homozygous variants were treated as single events. We assessed the association between each variant count category and ALS status using Firth's logistic regression with profile penalized likelihood CIs, with the 0-variant group as the reference category and adjusting for the same covariates used in the geneset burden analyses.

To identify nonrandom co-occurrence of variant pairs, we performed a permutation-based test within the case cohort. The set of variants tested included those defined above as well as *C9orf72* (C9) repeat expansion status (available for *n* = 8,610 cases; 66%). For each pair, we generated an empirical null distribution by performing 100,000 permutations, shuffling the carrier status of one variant relative to the other. An empirical *P* value was then calculated by comparing the observed co-occurrence count to this null distribution. For each variant pair, only participants with nonmissing genotypes for both variants were included.

Power analyses for the co-occurrence analyses were performed through 10,000 simulations. For each variant pair, joint genotype counts were drawn from a multinomial distribution, with probabilities based on their allele frequencies and a given co-occurrence OR. We then tested then for a depletion or excess of co-occurrence using Fisher's exact test comparing observed to expected counts under independence.

To test for statistical interactions among variant pairs, we used Firth's logistic regression in the full case–control cohort. For each pair of variants, we fitted a model including their main effects and their interaction term, adjusting for the same covariates as used in the single-variant analyses.

$$\text{MND} \sim \beta_0 + \beta_1 \times \text{var1} + \beta_2 \times \text{var2} + \beta_3 \times (\text{var1} \times \text{var2}) + \beta_4 \times \text{sex}$$
$$+ \beta_5 \times \text{total synonymous count} + \beta_6 \text{PC}_1 + \cdots + \beta_{15} \times \text{PC}_{10}$$

Because C9 status was available for only a few controls, it was not included in these case–control interaction models.

### Survival and age of onset analyses

Age at onset analyses (*n* = 10,557) were performed using linear regression, testing for an association between age at onset and either MAC (single-variant analyses) or the aggregate effect of minor alleles observed per sample per gene (URV burden analyses). Survival analyses (*n* = 7,194) were performed using a Cox proportional hazards model, testing for an association between right-censored survival time and either MAC (single-variant analyses) or the aggregate effect of minor alleles observed per sample per gene (URV burden analyses). Both age at onset and survival analyses were adjusted for sex, cohort, ten PCs and the total number of rare synonymous variants in each participant.

### Replication analyses

Power analyses were performed through 10,000 simulations in which alleles were drawn from the binomial distribution with the probability set to the MAF of the respective variant. Simulated genotypes were tested for an association with the binary phenotype status using Firth's logistic regression[12]. Power was calculated as the fraction of simulations with *P* values below the specified significance level. Effect sizes estimated in the discovery phase were corrected for winner's curse bias using the parametric bootstrap approach implemented in the winnerscurse R package[66]. Power analyses for URV burden tests were performed in a similar manner, substituting MAF for the frequency of carrying at least one minor allele across the gene. These power estimates reflect an idealized scenario without covariate adjustment; in practice, necessary covariate inclusion may reduce power.

The processing of sequencing data and sample quality control were performed identically to the discovery stage; participants who were duplicates or related up to the second degree to any participant in the discovery cohort were excluded. Single-variant analyses were performed identically to those in the discovery analyses and included variants that were significant in the discovery analysis (all variants achieved per-supercohort call-rate >0.9, HWE *P* value > 0.0001).

URV burden analyses were performed identically to those in the discovery analyses and included variants that passed strict quality control filters as applied in the discovery analysis. Meta-analyses were performed using Stouffer's $Z$ score method weighted by effective sample size, as implemented in METAL software[67].

## Gene annotation

Candidate genes were annotated with GO terms. The resulting list was summarized using the *rrvgo* R package[68], where a matrix of pairwise semantic similarity scores was first calculated. The terms were subsequently clustered using the default similarity threshold of 0.7 and a representative term for each cluster was selected based on its uniqueness score.

## Reporting summary

Further information on research design is available in the Nature Portfolio Reporting Summary linked to this article.

## Data availability

Project MinE data are available here: https://www.projectmine.com/research/data-sharing/. dbGAP datasets used are available under the following accession numbers: ALS compute (phs003184); Alzheimer's Disease Sequencing Project (ADSP) (phs000572); Autism Sequencing Consortium (ASC) (phs000298); Sweden-Schizophrenia Population-Based Case–Control Exome Sequencing (phs000473); Inflammatory Bowel Disease Exome Sequencing Study (phs001076); Myocardial Infarction Genetics Exome Sequencing Consortium: Ottawa Heart Study (phs000806); Myocardial Infarction Genetics Exome Sequencing Consortium: Malmo Diet and Cancer Study (phs001101); Myocardial Infarction Genetics Exome Sequencing Consortium: U. of Leicester (phs001000); Myocardial Infarction Genetics Exome Sequencing Consortium: Italian Atherosclerosis Thrombosis and Vascular Biology (phs000814); NHLBI GO-ESP: Women's Health Initiative Exome Sequencing Project (WHI)−WHISP (phs000281); Building on GWAS for NHLBI diseases: the US CHARGE Consortium (CHARGE-S): CHS (phs000667); Building on GWAS for NHLBI Diseases: the US CHARGE Consortium (CHARGE-S): ARIC (phs000668); Building on GWAS for NHLBI diseases: the US CHARGE consortium (CHARGE-S): FHS (phs000651); NHLBI GO-ESP Family Studies: Idiopathic Bronchiectasis (phs000518); NHLBI GO-ESP: Family Studies (Hematological Cancers) (phs000632); NHLBI GO-ESP: Family Studies: (familial atrial fibrillation) (phs000362); NHLBI GO-ESP: Heart Cohorts Exome Sequencing Project (ARIC) (phs000398); NHLBI GO-ESP: Heart Cohorts Exome Sequencing Project (CHS) (phs000400); NHLBI GO-ESP: Heart Cohorts Exome Sequencing Project (FHS) (phs000401); NHLBI GO-ESP: Lung Cohorts Exome Sequencing Project (asthma) (phs000422); NHLBI GO-ESP: Lung Cohorts Exome Sequencing Project (COPDGene) (phs000296); GO-ESP: Family Studies (Thoracic aortic aneurysms leading to acute aortic dissections) (phs000347). NHLBI TOPMed: Genomic Activities such as Whole Genome Sequencing and Related Phenotypes in the Framingham Heart Study (phs000974); NHLBI TOPMed: Genetics of Cardiometabolic Health in the Amish (phs000956); NHLBI TOPMed: Genetic Epidemiology of COPD (COPDGene) (phs000951); NHLBI TOPMed: The Vanderbilt Atrial Fibrillation Registry (VU_AF) (phs001032); NHLBI TOPMed: Cleveland Clinic Atrial Fibrillation (CCAF) Study (phs001189); NHLBI TOPMed: Partners HealthCare Biobank (phs001024); NHLBI TOPMed−NHGRI CCDG: Massachusetts General Hospital (MGH) Atrial Fibrillation Study (phs001062); NHLBI TOPMed: Novel Risk Factors for the Development of Atrial Fibrillation in Women (phs001040); NHLBI TOPMed−NHGRI CCDG: The Vanderbilt AF Ablation Registry (phs000997); NHLBI TOPMed: Heart and Vascular Health Study (HVH) (phs000993); NHLBI TOPMed−NHGRI CCDG: Atherosclerosis Risk in Communities (ARIC) (phs001211); NHLBI TOPMed: The Genetics and Epidemiology of Asthma in Barbados (phs001143); NHLBI TOPMed: Women's Health Initiative (WHI) (phs001237); NHLBI

TOPMed: Whole Genome Sequencing of Venous Thromboembolism (WGS of VTE) (phs001402); NHLBI TOPMed: Trans-Omics for Precision Medicine (TOPMed) Whole Genome Sequencing Project: Cardiovascular Health Study (phs001368). All participants gave written informed consent, and all studies were approved by the institutional review boards of the respective participating centers.

## Code availability

All raw sequencing data were aligned to the GRCh38 reference genome using BWA-mem (v.2.2.1) according to the pipeline described by Regier et al.[11] (implementation is available on GitHub at https://github.com/maarten-k/realignment and via Zenodo at https://doi.org/10.5281/zenodo.10963076 (ref. 69)). Joint genotyping was performed using a uniform pipeline according to the GATK best practices (v.4.2.6.1). Handling and filtering of VCF files was performed using VCFtools (v.0.1.16), BCFtools (v.1.9) and PLINK (v.1.90b6.21). Ancestry was estimated using LASER (v.2.04). Variants were annotated using Ensembl (GRCh38.105), snpEff (v.5.1d) and dbNSFP (v.4.3a). Sample and variant quality control was performed using PLINK (v.1.90b6.21) and RVAT (v.0.2.0), whereas sample relatedness was inferred using KING (v.2.2.7). Meta-analyses were performed using METAL (v.2011-03-25). GO terms were summarized using the rrvgo R package (v.1.18.0). All downstream analyses were performed using custom R code (performed in R v.3.6.3) that we made available in the RVAT R package (v.0.2.0) (available on GitHub at https://github.com/kennalab/rvat and via Zenodo at https://doi.org/10.5281/zenodo.10973472 (ref. 70)). Other R packages used either as dependencies of RVAT or in other analyses and visualizations are ggplot2 (v.3.4.2), ggrepel (v.0.9.1), dplyr (v.1.0.7), readr (v.2.1.1), stringr (v.1.4.0), tidyr (v.1.1.4), magrittr (v.2.0.1), kinship2 (v.1.9.6), logistf (v.1.25.0), SKAT (v.2.2.5), SummarizedExperiment (v.1.16.1), S4Vectors (v.0.24.4), GenomicRanges (v.1.38.0), IRanges (v.2.20.2), libraDBI (v.1.1.3), RSQLite (v.2.3.1), survival (v.3.1.8), winnerscurse (v.0.1.1). Figures were generated using R v.4.2.3, using rvat (v.0.3.4), dplyr (v.1.1.4), ggplot (v.3.5.1), readr (v.2.15), ggrepel (v.0.9.5), colorblindr (v.0.1.0), stringr (v.1.5.1), tidyr (v.1.3.1) and magrittr (v.2.0.3).

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

## Acknowledgements

This research has been conducted using the UK Biobank Resource under application number 48361. This project has received funding from the European Research Council under the European Union's Horizon 2020 research and innovation program (grant agreement no. 772376—EScORIAL. The collaboration project is cofunded by the PPP Allowance made available by Health–Holland, Top Sector Life Sciences and Health, to stimulate public–private partnerships. This study was supported by the ALS Foundation Netherlands. This work was sponsored by NWO–Domain Science for the use of supercomputer facilities. K.P.K. is supported by grants from the Dutch Research Council (grant no. ZonMW-VIDI 91719350) and the ALS Foundation Netherlands. Data from the New York Genome Center ALS consortium were used. All consortium members are listed in Supplementary Information. All New York Genome Center ALS Consortium activities are supported by the ALS Association (ALSA, grant no. 19-SI-459) and the Tow Foundation. R.L.M. receives support from the MND Association (grant no. 891-791). This publication has emanated from research conducted with the financial support of Taighde Éireann—Research Ireland, under grant no. 21/RC/10294_P2 at FutureNeuro Research Ireland Centre for Translational Brain Science. N.T. acknowledges the Italian Ministry of Health (grant no. RF-2021–12374238 'DRIVEALS') and the Italian Ministry of University and Research (Dipartimenti di Eccellenza Program 2023–2027—Department of Pathophysiology and Transplantation, Università degli Studi di Milano). E.R. is supported by the G. Harry Sheppard Memorial Research Fund and the ALS Society of Canada. Project MinE Belgium was supported by a grant from IWT (no. 140935), the ALS Liga België, the National Lottery of Belgium and the KU Leuven Opening the Future Fund. N.A.B. gratefully acknowledges the use of the services and facilities of Koc University-KUTTAM and the generous support of Suna and Inan Kirac Foundation. National Health and Medical Research Council of Australia, grant no. 1176913. M.A.v.E. and V.S. serve on the board of the European Reference Network (ERN) on neuromuscular disease (EURO–NMD). This publication has been supported by the ERN–NMD. F.C.G. was supported by the Scott Sullivan Fellowship from MND and Me and MND Research Australia. P.M.A. is supported by the Knut and Alice Wallenberg Foundation, the Fort Knox Charity Foundation, The Olsson and Olsson Foundation and the Swedish Brain Foundation. A.A.K. is funded by The Motor Neurone Disease Association (grant no. 1122462), NIHR Maudsley Biomedical Research Centre, ALS Association Milton Safenowitz Research Fellowship (grant no. RE19765), the Darby Rimmer MND Foundation, LifeArc (grant no. RE23378), MRC (grant no. MR/Z505705/1) and the Dementia Consortium (grant no. 1819242). A.A.K is supported by the UK Dementia Research Institute through UK DRI Ltd, funded principally by the Medical Research Council. A.A.-C. is an NIHR Senior Investigator (grant no. NIHR202421) and a Visiting Professor at the Perron Institute for Neurological and Translational Science, Australia. A.A.-C. is supported through the UK MND Research Institute (funders MND Association of England, Wales and Northern Ireland, My Name'5 Doddie Foundation, MND Scotland, LifeArc, MRC and NIHR) and through the Alan Davidson Foundation and Darby Rimmer Foundation. This study represents independent research part funded by the National Institute for Health Research (NIHR) Biomedical Research Centre at South London and Maudsley NHS Foundation Trust and King's College London. Samples used in this research were in part obtained from the UK National DNA Bank for MND Research, funded by the MND Association and the Wellcome Trust. We would like to thank people with MND and their families for their participation in this project. We acknowledge sample management undertaken by Biobanking Solutions funded by the Medical Research Council at the Centre for Integrated Genomic Medical Research, University of Manchester. This work was supported in part by the Intramural Research Program of the NIH, the National Institute on Aging (grant no. 1ZIAAG000933) and the National Institute of Neurological Disorders and Stroke (grant no. ZIANS003154). The contributions of the NIH authors were made as part of their official duties as NIH federal employees, are in compliance with agency policy requirements, and are considered Works of the United States Government. However, the findings and conclusions presented in this paper are those of the authors and do not necessarily reflect the views of NIH or the US Department of Health and Human Services.

## Author contributions

Data processing and analysis were performed by P.J.H., M.K., B.J.K., K.P.K. and J.H.V. Sample ascertainment and data generation was carried out by P.J.H., M.K., B.J.K., R.A.J.Z., K.R.V.E., Y.W., C.H.v.D., E.B., W.v.R., P.B., J.J.F.A.v.V., A.A.K., A.I., J.C.-K., B.N.S., S.T., A.J.v.d.K., V.F., V.D., Y.L., Y.S., D.B.R., K.L.W., R.L.M., J.H., Y.H., C.-Y.C., E.T., H.R., E.A., E.J.N.G., M.A.v.E., R.J.P., S.M.K.F., F.C.G., A.F.M., P.A.M., R.D.H., D.F., L.Š., H.H., A.L.N., R.J.C., L.B., B.R., B.K., J.Z., T.S., J.S.M.P., M.G., M.P., P. Corcia, P.V., P. Couratier, M.W., M.C.K., R.P., I.P.B., M.d.C., N.A.B., C.I., P.M.A., L.Z., E.R., I.R.M., N.D., G.A.R., B.J.T., N.T., A.C., V.S., O.H., H.P., M.B.H., C.L.D., J.D.G., J.E.L., P.V.D., K.E.M., P.J.S., C.E.S., A.A.-C., L.H.V.d.B., K.P.K. and

J.H.V. Writing of the paper was performed by P.J.H., K.P.K. and J.H.V. Study supervision was carried out by K.P.K. and J.H.V.

## Competing interests

J.H., Y.H., C.-Y.C., E.A.T. and H.R. are current or previous employees of Biogen. P.M.A. has served on advisory boards for Biogen, Regeneron, uniQure, Orphazyme A/S and Mitsubishi Pharma (mostly paid to institution). A.A.K. received consulting fees from the UK National Endowment for Science, Technology and the Arts (NESTA). N.T. received compensation for consulting services from Amylyx Pharmaceutical, Biogen, Italfarmaco and Zambon Biotech SA. V.S. received compensation for consulting services and/or speaking activities from AveXis, Cytokinetics, Italfarmaco, Liquidweb S.r.l., Novartis Pharma AG, Amylyx Pharmaceuticals, Biogen and Zambon Biotech SA, receives or has received research support from the Italian Ministry of Health, AriSLA and E-Rare Joint Transnational Call and is on the Editorial Board of *Amyotrophic Lateral Sclerosis and Frontotemporal Degeneration, European Neurology, American Journal of Neurodegenerative Diseases, Frontiers in Neurology* and *Exploration of Neuroprotective Therapy*. P.V.D. has served in advisory boards for Biogen, CSL Behring, Alexion Pharmaceuticals, Ferrer, QurAlis, Cytokinetics, argenx, UCB, Muna Therapeutics, Alector, Augustine Therapeutics, VectorY, Zambon, Amylyx, Novartis, Prilenia, Verge Genomics, Sapreme Technologies, Trace Neuroscience and NRG Therapeutics (paid to institution). P.V.D. has received speaker fees from Biogen and Amylyx (paid to institution). P.V.D. is supported by the E. von Behring Chair for Neuromuscular and Neurodegenerative Disorders (paid to institution). A.A.-C. reports consultancies or advisory boards for Amylyx, Apellis, Biogen, Clene Therapeutics, Cytokinetics, GenieUs, GSK, Lilly, Mitsubishi Tanabe Pharma, Novartis, OrionPharma, Quralis, Sano Genetics, Sanofi, Voyager Therapeutics and Wave Pharmaceuticals and the following patents or patent applications: WO2024121173A1 and EP 25 306 172.5. J.H.V. reports to have sponsored research agreements with Biogen, Eli Lilly, Trace and Astra Zeneca. The other authors declare no competing interests.

## Additional information

**Extended data** is available for this paper at https://doi.org/10.1038/s41588-026-02535-9.

**Correspondence and requests for materials** should be addressed to Kevin P. Kenna or Jan H. Veldink.

**a**

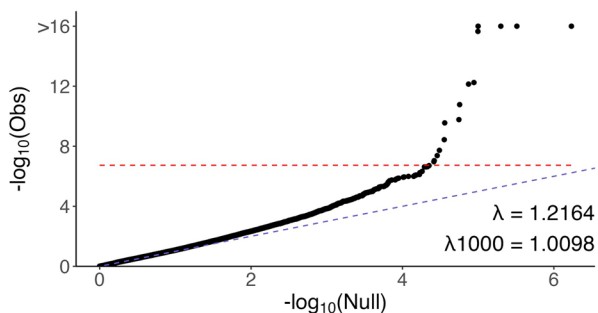

**b**

| variant | variant concordance | no case cohort bias | no allele imbalance | GWAS concordance |
|---|---|---|---|---|
| SOD1 (p.A5V) | ✔ | ✔ | ✔ | ✔ |
| SOD1 (p.D91A) | ✔ | ✔ | ✔ | ✔ |
| SOD1 (p.I114T) | ✔ | ✔ | ✔ | ✔ |
| CFAP410 (p.V58L) | ✔ | ✔ | ✔ | ✔ |
| NEK1 (p.S1036*) | ✔ | ✔ | ✔ | ✔ |
| NEK1 (p.R261H) | ✔ | ✔ | ✔ | ✔ |
| TBK1 (p.V464A) | ✔ | ✔ | ✔ | ✔ |
| KIF5A (p.P986L) | ✔ | ✔ | ✔ | ✔ |
| GREB1 (p.G1143D) | ✔ | ✔ | ✔ | ✗ |
| GBGT1 (p.R152L) | ✔ | ✔ | ✔ | ✔ |
| FUS (p.R521C) | ✔ | ✔ | ✔ | ✔ |
| CAPN2 (p.I530V) | ✔ | ✔ | ✔ | ✔ |
| SHCBP1 (p.M21T) | ✔ | ✔ | ✗ | ✔ |
| HTR3C (p.T186A) | ✔ | ✔ | ✔ | ✔ |
| TBK1 (p.K291E) | ✔ | ✔ | ✔ | ✔ |
| YKT6 (p.Y64C) | ✔ | ✔ | ✔ | ✔ |
| KNTC1 (p.W287R) | ✔ | ✔ | ✔ | ✔ |

**c**

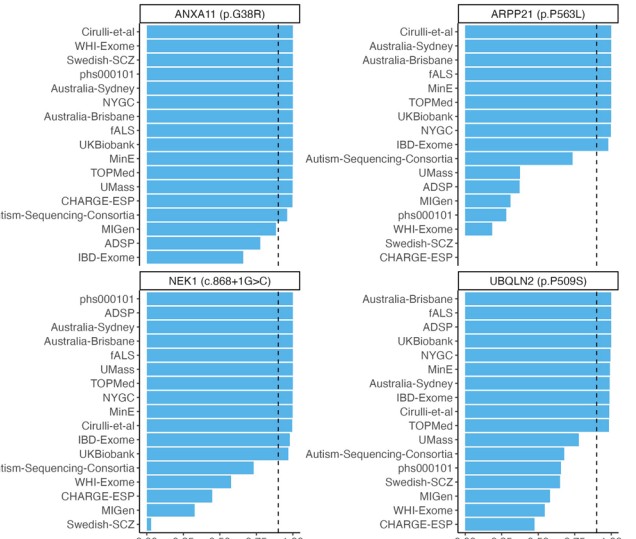

**d**

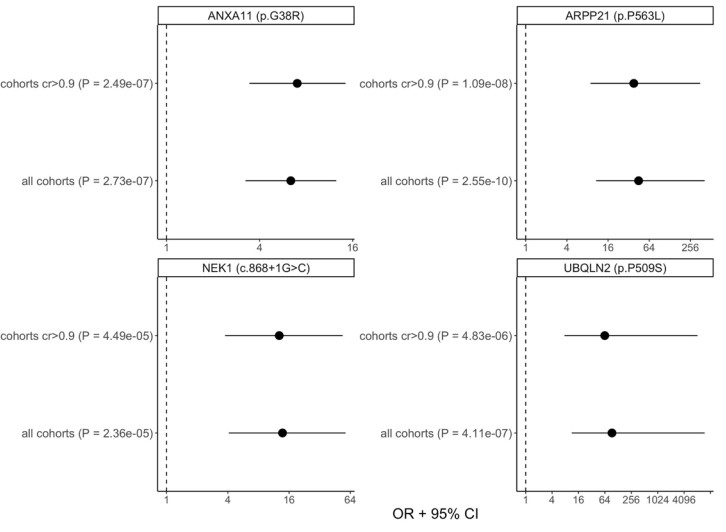

**Extended Data Fig. 1 | Rare single variant quantile-quantile plot and post-hoc analyses. a**, Quantile-quantile (qq) plot of observed single variant association $-\log_{10}$ (P-values) versus expected $-\log_{10}$ (P-values) under the null model. The red dotted line indicates the exome-wide significance threshold ($P = 1.83 \times 10^{-7}$). λ indicates the observed genomic inflation factor. $λ_{1000}$ indicates the genomic inflation factor for an equivalent study of 1,000 cases and 1,000 controls. **b**, Candidate single variant associations were screened for potential technical biases by assessing (i) variant concordance across 678 duplicates between cohorts present in the unfiltered dataset, (ii) biases among case cohorts using the same procedure as used in the control-control analyses, (iii) whether the minor

allele was supported a low number of reads, and (iv) significant heterogeneity ($P_{het} < 0.001$) between this study and the most recent ALS common variant GWAS by van Rheenen et al.[20] (for variants that overlap between both studies). **c**, Call-rates per cohort for the four variants in the targeted GCEP analysis that exhibited subpar call-rates. **d**, Forest plots comparing odds ratios (OR; center) with 95% confidence intervals (CI; error bars) for the variants shown in **c**. Results are displayed for analyses including all cohorts versus analyses restricted to cohorts meeting a call-rate threshold of 0.9. Association statistics were estimated using Firth's logistic regression with profile penalized likelihood confidence intervals. P-values are two-tailed and presented uncorrected for multiple testing.

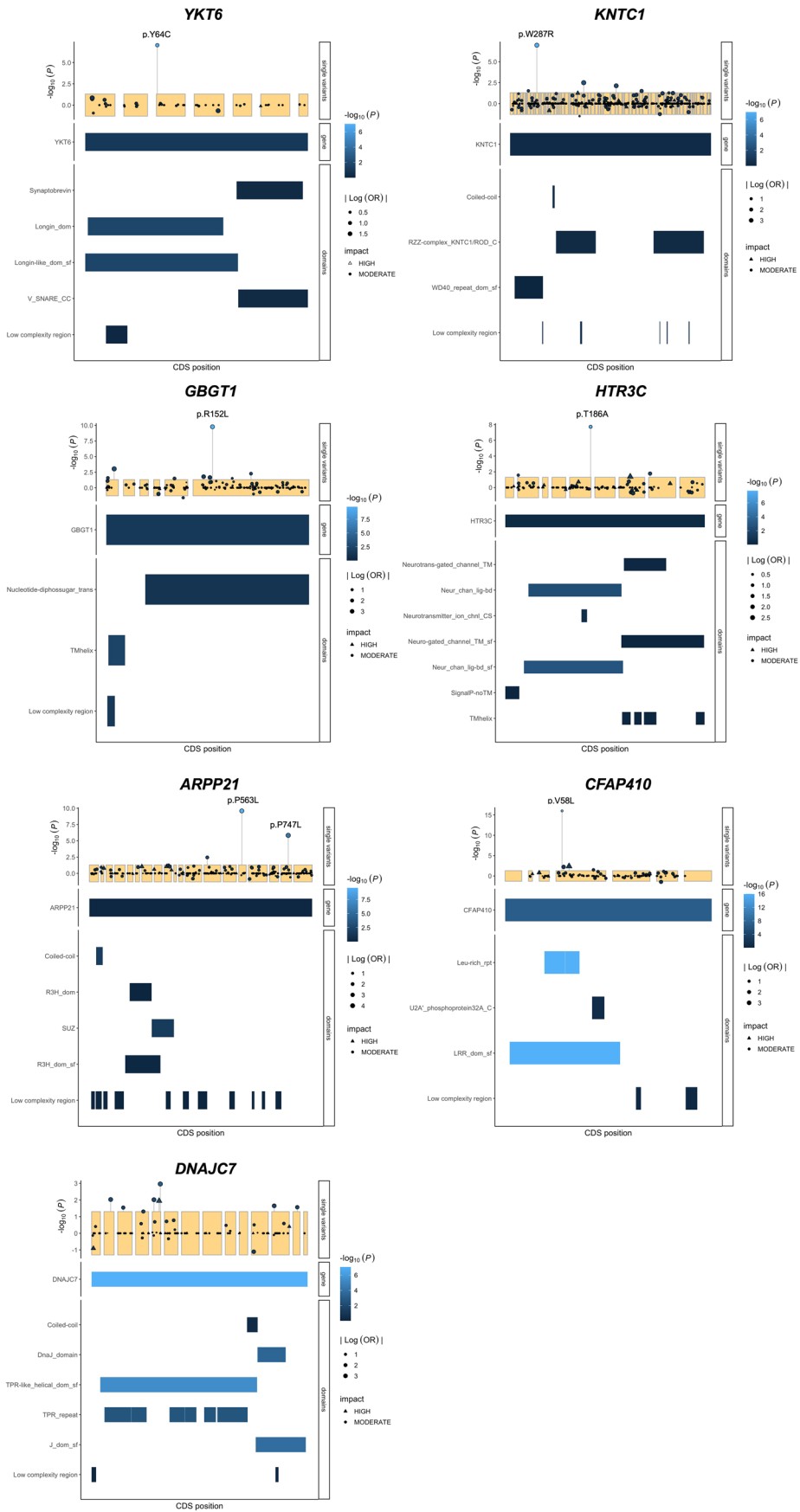

**Extended Data Fig. 2 | See next page for caption.**

**Extended Data Fig. 2 | Mutation plots for genes identified in rare single variant or URV burden analyses.** The upper panel shows the coding sequence of the respective genes, with the $y$-axis showing the $-\log_{10}(P$-value) for single variants. The lower panel shows the whole-gene domains colored by the $-\log_{10}(P$-value). Only variants and genes supported by replication are displayed. For variants identified in the rare single variant analysis, all variants with MAF < 0.05 are displayed; for genes identified in the URV burden analysis URVs are displayed (≤5 carriers). Association statistics were estimated using Firth's logistic regression with profile penalized likelihood confidence intervals. $P$-values for the gene-based and domain-based tests are from the ACAT omnibus test combining the four variant filtering strategies (see Methods). $P$-values shown are from two-tailed tests and are presented uncorrected for multiple testing.

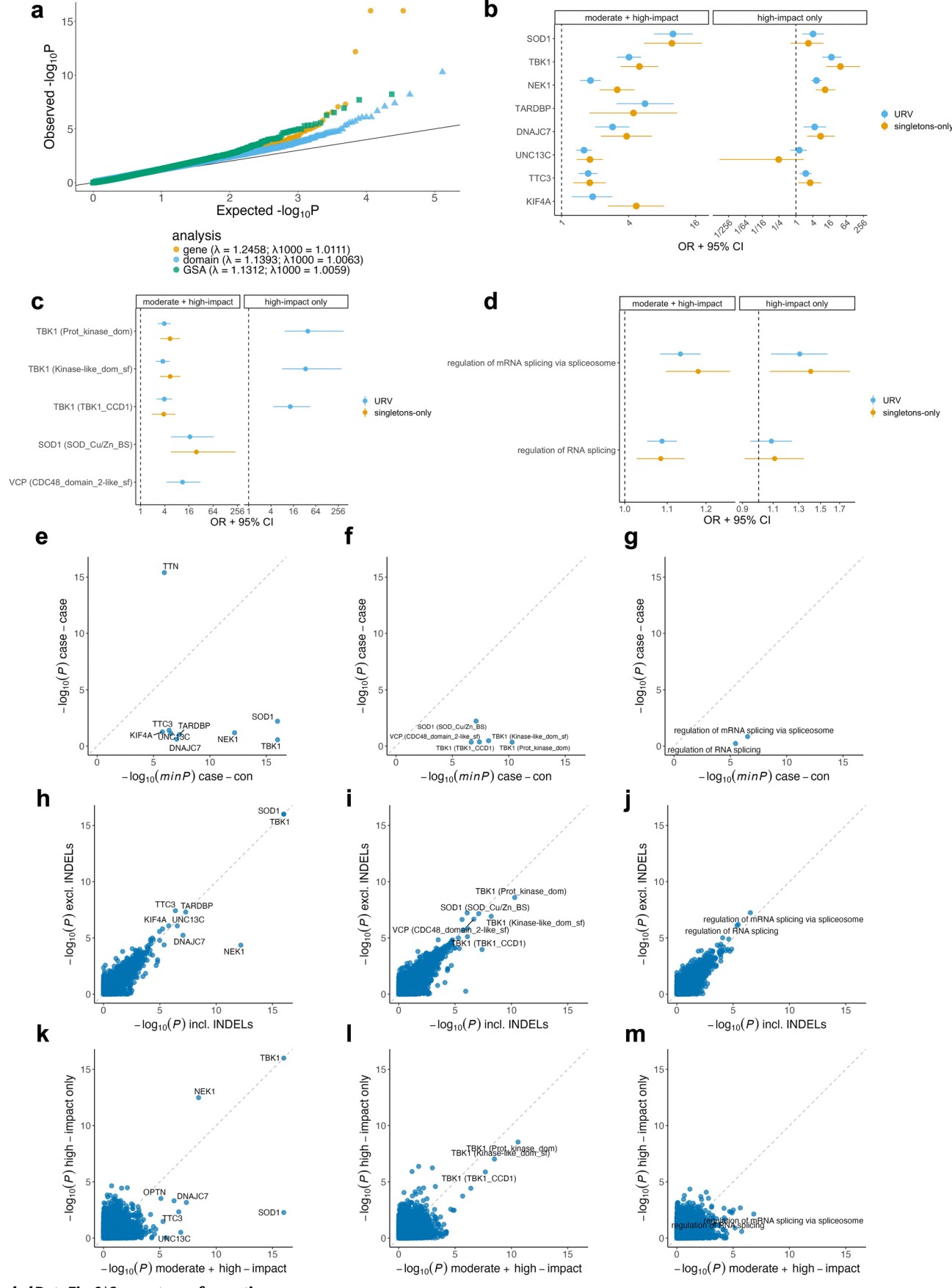

**Extended Data Fig. 3 | See next page for caption.**

**Extended Data Fig. 3 | Ultra-rare variant burden analyses. a**, Quantile-quantile (qq) plot of observed association $-\log_{10}$ ($P$-values) versus expected $-\log_{10}$ ($P$-values) under the null model for gene, domain and gene set analyses. **b–d**, Forest plots depicting odds-ratios (center) and 95% confidence intervals (error bars) of the exome-wide significant genes (**b**), domains (**c**) and gene sets (**d**). **e–g**, Significant associations were screened for technical biases arising from the inclusion of multiple case cohorts by testing for an association with cohort among ALS cases (**e**, gene-based analyses; **f**, domain-based analyses, **g**, gene-set analyses). **h–j**, Comparison of association $P$-values of ultra-rare gene burden analyses excluding INDELs ($y$-axis) compared to the main analysis including INDELs ($x$-axis) (**h**, gene-based analyses; **i**, domain-based analyses; **j**, gene-set analyses). **k–m**, Comparison of association $P$-values of ultra-rare gene burden analyses including high-impact variants only ($y$-axis) compared to including both moderate-impact and high-impact variants ($x$-axis) (**k**, gene-based analyses; **l**, domain-based analyses; **m**, gene-set analyses). Association statistics were estimated using Firth's logistic regression with profile penalized likelihood confidence intervals. $P$-values for the gene-based, domain-based and gene set analyses are from the ACAT omnibus test combining the four variant filtering strategies (see Methods). $P$-values are two-tailed and are presented uncorrected for multiple testing.

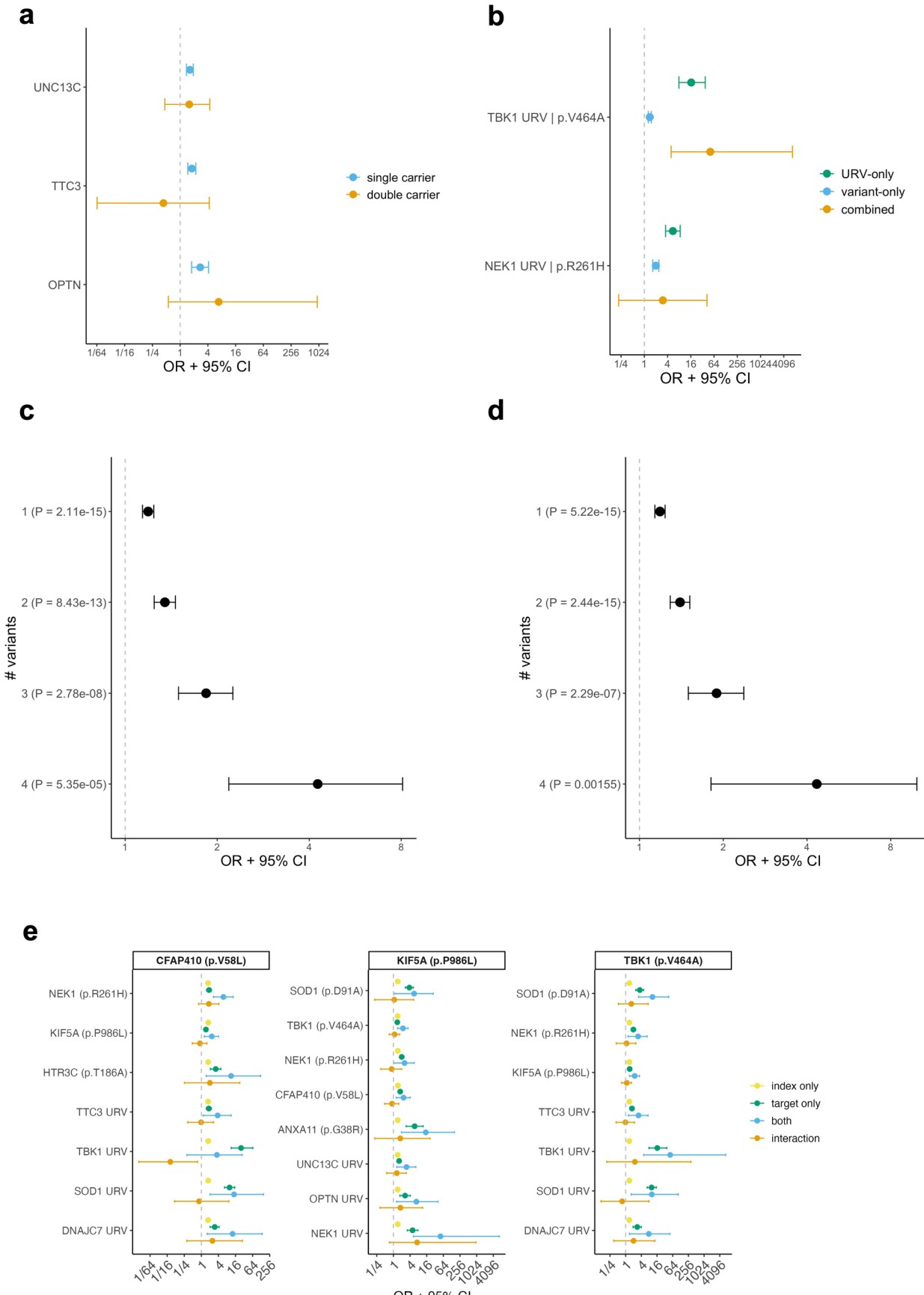

**Extended Data Fig. 4 | See next page for caption.**

**Extended Data Fig. 4 | Co-occurrence analyses. a, b,** Within-gene co-occurrence among genes identified in the ultra-rare variant burden analyses. The forest plots show the odds-ratio (OR; center) and 95% confidence intervals (CI; error bars), comparing the risk for individuals carrying a single qualifying variant to those carrying multiple variants. The panels depict genes in which co-occurrence of ultra-rare variants was observed (**a**), as well as genes in which co-occurrence of an ultra-rare variant with a more common variant from the single variant analyses was observed (**b**). **c, d,** Cumulative burden of carrying multiple high- or moderate-impact variants among Definitive ALS genes as curated by the GCEP. Shown are the odds-ratios (OR; center) and 95% confidence intervals (CI; error bars) (x-axis), stratified by the number of risk variants carried (y-axis). **c,** Analysis at the variant level, where each qualifying variant is counted individually. **d,** Analysis at the gene level, where carrying one or more qualifying variants within the same gene is counted as a single event. **e,** Forest plots showing the odds ratios (OR; center) and 95% confidence intervals (CI; error bars) for variant pairs. For each low-frequency variant (MAF 0.01-0.05), the odds-ratio is shown for carrying only the index variant (yellow, panel), only the target variant (green, y-axis), and carrying both variants (blue). The interaction odds-ratio (orange) indicates whether the variants act synergistically (OR > 1) or antagonistically (OR < 1). Association statistics are two-tailed and were estimated using Firth's logistic regression with profile penalized likelihood confidence intervals. P-values are two-tailed and presented uncorrected for multiple testing.

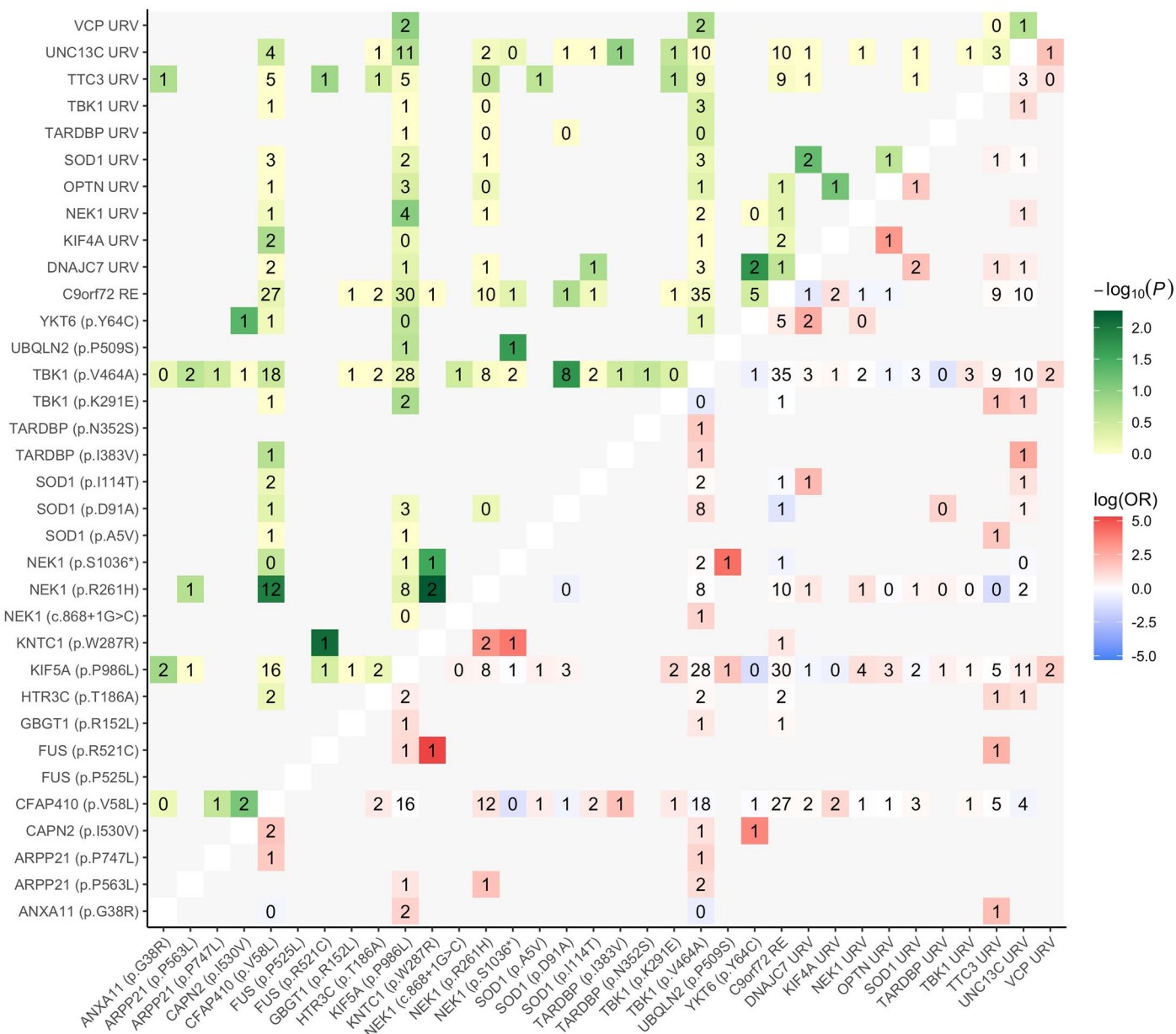

**Extended Data Fig. 5 | Variant co-occurrence heatmap.** Heatmap displaying the observed co-occurrences in ALS patients among variant pairs. For genes identified in the URV burden analysis, an individual is considered a carrier if they had at least one qualifying variant in that gene. The lower triangle shows the log-transformed odds ratio, where red indicates a synergistic effect (greater co-occurrence than expected under an additive risk model), and blue indicates an antagonistic effect (lesser co-occurrence than expected under an additive risk model). The upper triangle shows the statistical significance from a permutation analysis ($-\log_{10}(P$-value)), with darker green indicating a more significant $P$-value. Pairs for which there was no co-occurrence in the full dataset (cases and controls) are greyed out. In addition to variants identified in this study, we included *C9orf72* repeat expansion status in these co-occurrence analyses, which was available for 8,610 (66%) of cases. $P$-values are two-tailed and presented uncorrected for multiple testing.

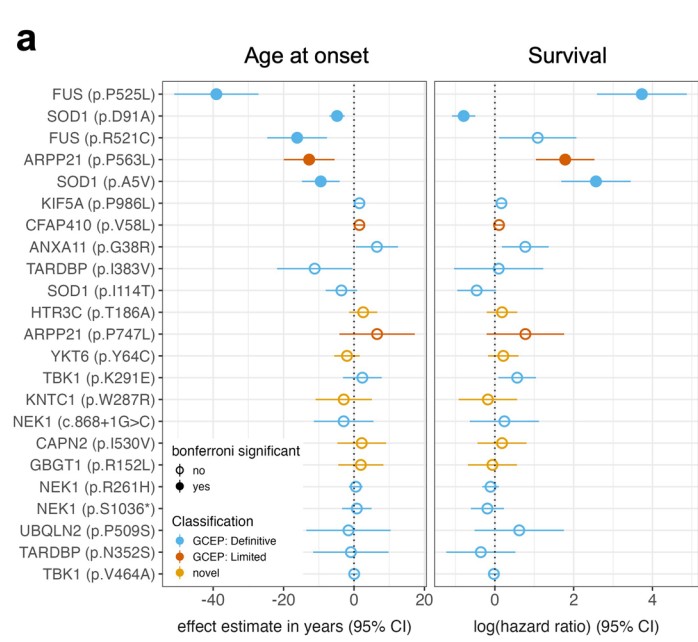

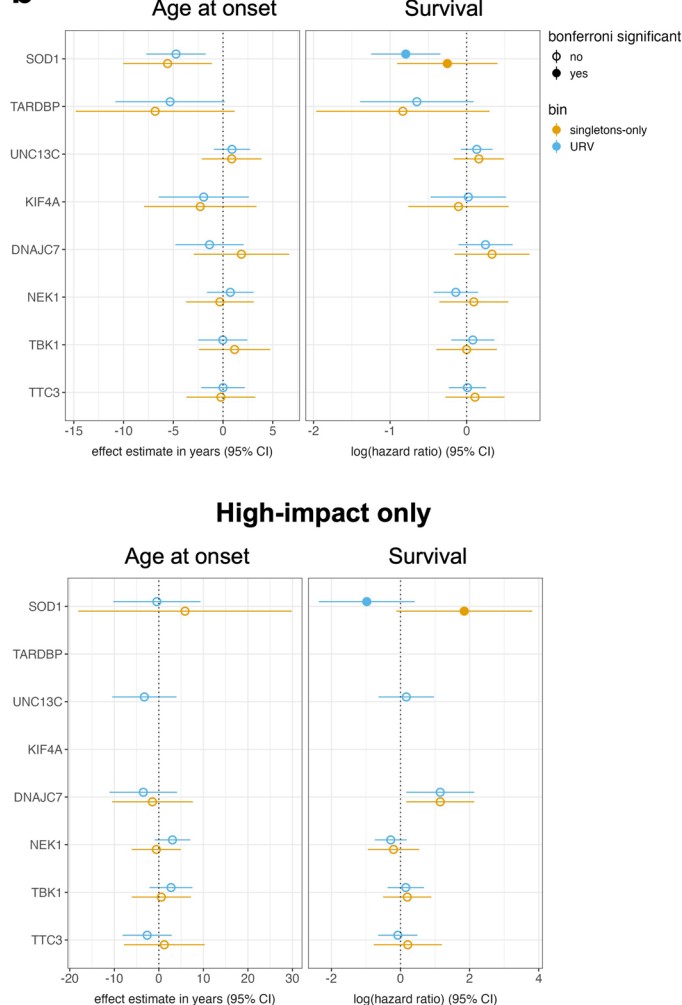

**Extended Data Fig. 6 | Age at onset and survival analyses. a, b,** Age at onset analyses (left) and survival analyses (right) for significant variants (**a**) and significant genes among URV burden analyses (**b**). Estimates in years (center) and 95% confidence intervals (error bars) are shown for age at onset analyses in 10,557 participants (linear model), and log-transformed hazard ratios (center) and 95% confidence intervals (error bars) are shown for survival analyses in 7,194 participants (Cox proportional hazards model). Closed circles indicate variants or genes that reached significance corrected for the number of variants or genes tested.

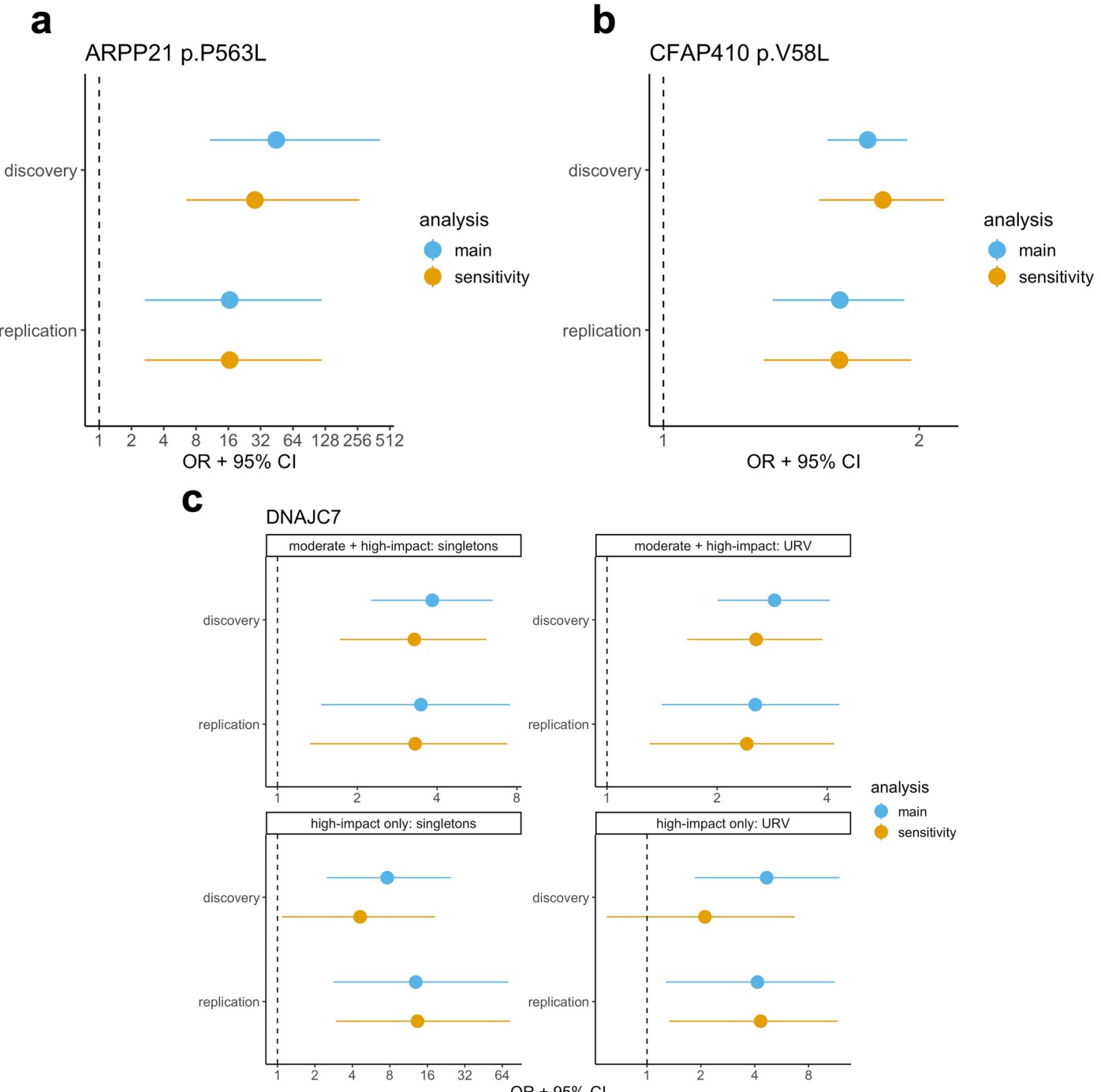

**Extended Data Fig. 7 | Independent validation of genes with limited prior evidence.** Forest plots comparing odds ratios (OR; center) with 95% confidence intervals (CI; error bars) for main analyses versus sensitivity analyses that exclude potential overlap with prior studies. **a**, Plot comparing the *ARPP21* p.P563L association from the full dataset to an analysis excluding four carriers who could potentially overlap with previous family studies. **b**, Plot comparing the *CFAP410* p.V58L association in the full discovery ($n_{cases} = 11{,}763$; $n_{controls} = 69{,}677$) and replication ($n_{cases} = 4{,}781$; $n_{controls} = 130{,}927$) cohorts to a sensitivity analysis that excludes duplicated or genetically related individuals from the discovery

($n_{cases} = 4{,}474$; $n_{controls} = 65{,}643$) and replication ($n_{cases} = 3{,}698$; $n_{control} = 130{,}802$) sets. **c**, Plot comparing the *DNAJC7* URV association in the full discovery ($n_{cases} = 13{,}138$; $n_{controls} = 69{,}775$) and replication ($n_{cases} = 4{,}781$; $n_{controls} = 130{,}928$) cohorts to a sensitivity analysis that excludes individuals from cohorts that potentially overlap with the original study from the discovery ($n_{cases} = 7{,}606$; $n_{controls} = 59{,}926$) and replication ($n_{cases} = 4{,}591$; $n_{controls} = 130{,}928$) sets. Association statistics were estimated using Firth's logistic regression with profile penalized likelihood confidence intervals.

## Extended Data Table 1 | Rare single variants in GCEP-curated genes achieving significance

| | | | | discovery | | | | replication | | | | meta-analysis |
|---|---|---|---|---|---|---|---|---|---|---|---|---|
| Variant | Gene | Consequence | GCEP classification | case MAC (MAF) | ctrl MAC (MAF) | OR (95% CI) | P | case MAC (MAF) | ctrl MAC (MAF) | OR (95% CI) | P | P |
| 21:31667359:T:C | SOD1 | c.341T>C/p.I114T | Definitive | 41 ($1.56 \times 10^{-3}$) | 1 ($7.17 \times 10^{-6}$) | 175 (46.8–1557) | $< 1 \times 10^{-16}$ | 11 ($1.15 \times 10^{-3}$) | 11 ($4.20 \times 10^{-5}$) | 135 (50.7–351) | $2.22 \times 10^{-16}$ | $< 1 \times 10^{-16}$ |
| 21:44333234:C:A | CFAP410 | c.172G>T/p.V58L | Limited | 451 ($1.92 \times 10^{-2}$) | 1620 ($1.16 \times 10^{-2}$) | 1.74 (1.56–1.94) | $< 1 \times 10^{-16}$ | 148 ($1.55 \times 10^{-2}$) | 3056 ($1.17 \times 10^{-2}$) | 1.61 (1.34–1.92) | $6.83 \times 10^{-7}$ | $< 1 \times 10^{-16}$ |
| 21:31667290:A:C | SOD1 | c.272A>C/p.D91A | Definitive | 94 ($3.58 \times 10^{-3}$) | 95 ($6.81 \times 10^{-4}$) | 3.63 (2.74–4.83) | $< 1 \times 10^{-16}$ | 26 ($2.72 \times 10^{-3}$) | 180 ($6.87 \times 10^{-4}$) | 3.32 (2.10–5.07) | $1.70 \times 10^{-6}$ | $< 1 \times 10^{-16}$ |
| 4:169424668:G:C | NEK1 | c.3107C>G/p.S1036* | Definitive | 42 ($1.60 \times 10^{-3}$) | 39 ($2.79 \times 10^{-4}$) | 7.02 (4.51–10.9) | $2.22 \times 10^{-16}$ | 12 ($1.25 \times 10^{-3}$) | 78 ($2.98 \times 10^{-4}$) | 5.06 (2.51–9.35) | $3.60 \times 10^{-5}$ | $< 1 \times 10^{-16}$ |
| 21:31659783:C:T | SOD1 | c.14C>T/p.A5V | Definitive | 24 ($9.14 \times 10^{-4}$) | 0 (0.00) | 262 (36.2–33313) | $< 1 \times 10^{-16}$ | 3 ($3.14 \times 10^{-4}$) | 0 (0.00) | 189 (15.3–26134) | $1.04 \times 10^{-4}$ | $< 1 \times 10^{-16}$ |
| 4:169585374:C:T | NEK1 | c.782G>A/p.R261H | Definitive | 174 ($6.63 \times 10^{-3}$) | 437 ($3.13 \times 10^{-3}$) | 2.01 (1.68–2.41) | $5.66 \times 10^{-13}$ | 72 ($7.53 \times 10^{-3}$) | 752 ($2.87 \times 10^{-3}$) | 2.20 (1.65–2.89) | $2.90 \times 10^{-7}$ | $< 1 \times 10^{-16}$ |
| 12:57581917:C:T | KIF5A | c.2957C>T/p.P986L | Definitive | 497 ($1.89 \times 10^{-2}$) | 1932 ($1.38 \times 10^{-2}$) | 1.44 (1.30–1.59) | $1.68 \times 10^{-11}$ | 172 ($1.80 \times 10^{-2}$) | 3509 ($1.34 \times 10^{-2}$) | 1.57 (1.33–1.85) | $3.92 \times 10^{-7}$ | $< 1 \times 10^{-16}$ |
| 12:64488537:T:C | TBK1 | c.1391T>C/p.V464A | Definitive | 688 ($2.62 \times 10^{-2}$) | 2996 ($2.15 \times 10^{-2}$) | 1.38 (1.27–1.50) | $7.24 \times 10^{-13}$ | 210 ($2.20 \times 10^{-2}$) | 6101 ($2.33 \times 10^{-2}$) | 1.24 (1.07–1.43) | $5.02 \times 10^{-3}$ | $4.38 \times 10^{-14}$ |
| 1:11022464:A:G | TARDBP | c.1055A>G/p.N352S | Definitive | 6 ($2.30 \times 10^{-4}$) | 0 (0.00) | 67.7 (7.82–8874) | $8.78 \times 10^{-6}$ | 10 ($1.05 \times 10^{-3}$) | 0 (0.00) | 173 (21.1–22528) | $9.43 \times 10^{-10}$ | $1.69 \times 10^{-12}$ |
| 3:35738257:C:T | ARPP21 | c.1688C>T/p.P563L | Limited | 18 ($7.02 \times 10^{-4}$) | 1 ($8.44 \times 10^{-6}$) | 44.8 (10.7–414) | $2.55 \times 10^{-10}$ | 3 ($3.14 \times 10^{-4}$) | 2 ($7.64 \times 10^{-6}$) | 16.5 (2.66–118) | $3.29 \times 10^{-3}$ | $5.35 \times 10^{-12}$ |
| 16:31191418:C:T | FUS | c.1561C>T/p.R521C | Definitive | 13 ($5.18 \times 10^{-4}$) | 1 ($7.20 \times 10^{-6}$) | 53.0 (12.6–490) | $2.78 \times 10^{-10}$ | 1 ($1.05 \times 10^{-4}$) | 0 (0.00) | 56.7 (3.02–8273) | $9.29 \times 10^{-3}$ | $2.03 \times 10^{-11}$ |
| 12:64481900:A:G | TBK1 | c.871A>G/p.K291E | Definitive | 20 ($7.62 \times 10^{-4}$) | 21 ($1.51 \times 10^{-4}$) | 6.21 (3.34–11.5) | $4.20 \times 10^{-8}$ | 6 ($6.27 \times 10^{-4}$) | 42 ($1.60 \times 10^{-4}$) | 6.00 (2.34–13.1) | $6.82 \times 10^{-4}$ | $1.13 \times 10^{-10}$ |
| 1:11022556:A:G | TARDBP | c.1147A>G/p.I383V | Definitive | 8 ($3.07 \times 10^{-4}$) | 1 ($7.23 \times 10^{-6}$) | 22.9 (4.99–219) | $1.54 \times 10^{-5}$ | 4 ($4.18 \times 10^{-4}$) | 2 ($7.64 \times 10^{-6}$) | 39.0 (7.10–290) | $4.77 \times 10^{-5}$ | $5.22 \times 10^{-9}$ |
| 3:35792484:C:T | ARPP21 | c.2240C>T/p.P747L | Limited | 8 ($3.05 \times 10^{-4}$) | 0 (0.00) | 75.8 (9.06–9872) | $1.45 \times 10^{-6}$ | 3 ($3.14 \times 10^{-4}$) | 1 ($3.82 \times 10^{-6}$) | 31.3 (3.97–370) | $1.19 \times 10^{-3}$ | $6.43 \times 10^{-9}$ |
| 10:80170859:C:T | ANXA11 | c.112C>A/p.G38R | Definitive | 18 ($6.85 \times 10^{-4}$) | 17 ($1.26 \times 10^{-4}$) | 6.37 (3.24–12.5) | $2.73 \times 10^{-7}$ | 2 ($2.09 \times 10^{-4}$) | 25 ($9.55 \times 10^{-5}$) | 4.80 (0.942–15.6) | $5.73 \times 10^{-2}$ | $8.85 \times 10^{-8}$ |
| 4:169580841:C:G | NEK1 | c.868+1G>C | Definitive | 8 ($3.05 \times 10^{-4}$) | 3 ($2.48 \times 10^{-5}$) | 13.8 (4.08–57.5) | $2.36 \times 10^{-5}$ | 3 ($3.14 \times 10^{-4}$) | 14 ($5.35 \times 10^{-5}$) | 8.26 (2.04–25.4) | $5.59 \times 10^{-3}$ | $4.32 \times 10^{-7}$ |
| 16:31191431:C:T | FUS | c.1574C>T/p.P525L | Definitive | 8 ($3.06 \times 10^{-4}$) | 0 (0.00) | 89.9 (10.7–11716) | $4.57 \times 10^{-7}$ | 1 ($1.05 \times 10^{-4}$) | 0 (0.00) | 4.56 (0.243–665) | $3.11 \times 10^{-1}$ | $1.70 \times 10^{-6}$ |
| X:56565398:C:T | UBQLN2 | c.1525C>T/p.P509S | Definitive | 7 ($3.85 \times 10^{-4}$) | 0 (0.00) | 92.8 (11.2–12067) | $4.11 \times 10^{-7}$ | 0 (0.00) | 1 ($4.95 \times 10^{-6}$) | 5.28 (0.0361–99.1) | $3.84 \times 10^{-1}$ | $2.40 \times 10^{-6}$ |

Listed are variants that reached significance in the GCEP-targeted analysis ($P < 3.20 \times 10^{-5}$). Test statistics are shown for the discovery phase ($n_{cases} = 13{,}138$; $n_{controls} = 69{,}775$), replication phase ($n_{cases} = 4{,}781$; $n_{controls} = 130{,}928$), and the combined meta-analysis (Stouffer's Z-score method, weighted by effective sample size). Association statistics were estimated using Firth's logistic regression with profile penalized likelihood confidence intervals. P-values are two-tailed and presented uncorrected for multiple testing.

**Extended Data Table 2 | Genes achieving significance among GCEP-curated genes in URV burden analyses**

| Gene | GCEP classification | Filtering strategy | discovery | | | | replication | | | | meta-analysis |
|------|--------------------|--------------------|-----------|---|---|---|-------------|---|---|---|---------------|
| | | | # case carriers (frequency) | # ctrl carriers (frequency) | OR (95% CI) | P | # case carriers (frequency) | # ctrl carriers (frequency) | OR (95% CI) | P | P |
| SOD1 | Definitive | moderate-impact / ultra-rare | 85 ($6.47 \times 10^{-3}$) | 38 ($5.46 \times 10^{-4}$) | 10.0 (6.78-15.0) | $< 1 \times 10^{-16}$ | 38 ($7.95 \times 10^{-3}$) | 56 ($4.28 \times 10^{-4}$) | 13.1 (7.71-21.8) | $< 1.00 \times 10^{-16}$ | $< 1 \times 10^{-16}$ |
| TBK1 | Definitive | high-impact / ultra-rare | 33 ($2.51 \times 10^{-3}$) | 8 ($1.18 \times 10^{-4}$) | 18.5 (8.98-42.4) | $< 1 \times 10^{-16}$ | 7 ($1.46 \times 10^{-3}$) | 31 ($2.37 \times 10^{-4}$) | 5.93 (2.10-14.4) | $2.14 \times 10^{-10}$ | $< 1 \times 10^{-16}$ |
| NEK1 | Definitive | high-impact / ultra-rare | 47 ($3.58 \times 10^{-3}$) | 43 ($6.21 \times 10^{-4}$) | 5.38 (3.50-8.29) | $6.49 \times 10^{-13}$ | 22 ($4.60 \times 10^{-3}$) | 98 ($7.49 \times 10^{-4}$) | 6.26 (3.53-10.6) | $4.33 \times 10^{-8}$ | $< 1 \times 10^{-16}$ |
| TARDBP | Definitive | moderate-impact / ultra-rare | 25 ($1.91 \times 10^{-3}$) | 22 ($3.16 \times 10^{-4}$) | 5.61 (3.12-10.1) | $5.02 \times 10^{-8}$ | 17 ($3.56 \times 10^{-3}$) | 106 ($8.10 \times 10^{-4}$) | 3.53 (1.76-6.57) | $1.29 \times 10^{-3}$ | $1.29 \times 10^{-10}$ |
| DNAJC7 | Limited | moderate-impact / ultra-rare | 58 ($4.42 \times 10^{-3}$) | 86 ($1.24 \times 10^{-3}$) | 2.87 (2.01-4.07) | $8.77 \times 10^{-8}$ | 19 ($4.18 \times 10^{-3}$) | 153 ($1.17 \times 10^{-3}$) | 2.54 (1.41-4.32) | $2.63 \times 10^{-3}$ | $9.26 \times 10^{-10}$ |
| OPTN | Definitive | moderate-impact / singletons-only | 42 ($3.35 \times 10^{-3}$) | 59 ($8.46 \times 10^{-4}$) | 2.70 (1.78-4.07) | $1.56 \times 10^{-5}$ | 11 ($2.30 \times 10^{-3}$) | 51 ($3.97 \times 10^{-4}$) | 1.76 (0.656-4.48) | $3.69 \times 10^{-1}$ | $2.89 \times 10^{-5}$ |

Listed are genes that reached significance in a targeted analysis among GCEP-curated in the URV burden analyses ($P < 1.0 \times 10^{-3}$). Test statistics are shown for the discovery phase ($n_{cases} = 13{,}138$; $n_{controls} = 69{,}775$), replication phase ($n_{cases} = 4{,}781$; $n_{controls} = 130{,}928$), and the combined meta-analysis (Stouffer's $Z$-score method, weighted by effective sample size). Carrier frequencies, odds ratios (OR), and confidence intervals (CI) estimated using Firth's logistic regression with profile penalized likelihood confidence intervals are presented for the most significant of the four variant filtering strategies. $P$-values are two-tailed, uncorrected for multiple testing, and estimated using the ACAT omnibus test combining the four variant filtering strategies.

**Extended Data Table 3 | Cumulative percentage of patients carrying ALS-associated variants**

| | # variants | | | |
|---|---|---|---|---|
| **variants included** | **≥1** | **1** | **2** | **3** |
| GCEP 'definitive' (sv) | 11.6 | 11.1 | 0.540 | 0 |
| + GCEP 'limited' (sv) | 14.8 | 13.8 | 0.959 | 0.00761 |
| + novel (sv) | 15.6 | 14.5 | 1.10 | 0.00761 |
| + GCEP 'definitive' (URV) | 17.2 | 15.9 | 1.32 | 0.0152 |
| + GCEP 'limited' (URV) | 17.6 | 16.1 | 1.42 | 0.0152 |
| + novel (URV) | 20.0 | 18.2 | 1.70 | 0.0989 |
| + C9 RE* | 26.9 | 23.5 | 3.12 | 0.221 |

The table shows the percentage of ALS patients carrying at least one (≥1), exactly one (1), two (2), or three (3) qualifying variants identified in this study. Each row represents the cumulative proportion of patients relative to the row above it, starting with single variants (sv) and adding qualifying variants (URV) in genes identified in the burden analyses. *Data on the *C9orf72* (C9) repeat expansion was available for 66% of patients. When only single variants + C9 status are considered, the percentages are 22.9% (≥1), 20.5% (1), 2.36% (2) and 0.09% (3), respectively. sv, single variants; URV, ultra-rare variants.

# Reporting Summary

## Statistics

For all statistical analyses, confirm that the following items are present in the figure legend, table legend, main text, or Methods section.

| n/a | Confirmed | |
|---|---|---|
| ☐ | ☒ | The exact sample size (*n*) for each experimental group/condition, given as a discrete number and unit of measurement |
| ☒ | ☐ | A statement on whether measurements were taken from distinct samples or whether the same sample was measured repeatedly |
| ☐ | ☒ | The statistical test(s) used AND whether they are one- or two-sided<br>*Only common tests should be described solely by name; describe more complex techniques in the Methods section.* |
| ☐ | ☒ | A description of all covariates tested |
| ☐ | ☒ | A description of any assumptions or corrections, such as tests of normality and adjustment for multiple comparisons |
| ☐ | ☒ | A full description of the statistical parameters including central tendency (e.g. means) or other basic estimates (e.g. regression coefficient) AND variation (e.g. standard deviation) or associated estimates of uncertainty (e.g. confidence intervals) |
| ☐ | ☒ | For null hypothesis testing, the test statistic (e.g. *F*, *t*, *r*) with confidence intervals, effect sizes, degrees of freedom and *P* value noted<br>*Give P values as exact values whenever suitable.* |
| ☒ | ☐ | For Bayesian analysis, information on the choice of priors and Markov chain Monte Carlo settings |
| ☒ | ☐ | For hierarchical and complex designs, identification of the appropriate level for tests and full reporting of outcomes |
| ☐ | ☒ | Estimates of effect sizes (e.g. Cohen's *d*, Pearson's *r*), indicating how they were calculated |

*Our web collection on statistics for biologists contains articles on many of the points above.*

## Software and code

Policy information about availability of computer code

Data collection | All raw sequencing data was aligned to the GRCh38 reference genome using BWA-mem (v.2.2.1) according to the pipeline described by Regier et al. (implementation can be found at GitHub (https://github.com/maarten-k/realignment) and Zenodo (https://doi.org/10.5281/zenodo.10963076). Joint genotyping was performed using a uniform pipeline according to the GATK best practices (v. 4.2.6.1).

Data analysis | All raw sequencing data was aligned to the GRCh38 reference genome using BWA-mem (v.2.2.1) according to the pipeline described by Regier et al. (implementation can be found at GitHub (https://github.com/maarten-k/realignment) and Zenodo (https://doi.org/10.5281/zenodo.10963076). Joint genotyping was performed using a uniform pipeline according to the GATK best practices (v. 4.2.6.1). Handling and filtering of VCF files was performed using VCFtools (v. 0.1.16), BCFtools (v. 1.9) and PLINK (v. 1.90b6.21). Ancestry was estimated using LASER (v.2.04). Variants were annotated using Ensembl (GRCh38.105), snpEff (v.5.1d) and dbNSFP (v. 4.3a). Sample and variant quality control was performed using PLINK (v. 1.90b6.21) and RVAT (v. 0.2.0), while sample relatedness was inferred using KING (v. 2.2.7). Meta-analyses were performed using METAL (v. 2011-03-25). Gene ontology terms were summarized using the rrvgo R package (v. 1.18.0). All downstream analyses were performed using custom R code (performed in R 3.6.3) that we made available in the RVAT R package (v. 0.2.0) (available on GitHub: https://github.com/kennalab/rvat and Zenodo: https://doi.org/10.5281/zenodo.10973472). Other R packages used either as dependencies of RVAT or in other analyses and visualizations are ggplot2 (v. 3.4.2), ggrepel (v. 0.9.1), dplyr (v. 1.0.7), readr (v. 2.1.1), stringr (v. 1.4.0), tidyr (v. 1.1.4), magrittr (v. 2.0.1), kinship2 (v. 1.9.6), logistf (v. 1.25.0), SKAT (v. 2.2.5), SummarizedExperiment (v. 1.16.1), S4Vectors (v. 0.24.4), GenomicRanges (v. 1.38.0), IRanges (v. 2.20.2), libraDBI (v. 1.1.3), RSQLite (v. 2.3.1), survival (v. 3.1.8), winnerscurse (v. 0.1.1). Figures were generated using R 4.2.3, using rvat (v. 0.3.4), dplyr (v.1.1.4), ggplot (v. 3.5.1), readr (v. 2.15), ggrepel (v. 0.9.5), colorblindr (v. 0.1.0), stringr (v. 1.5.1), tidyr (v. 1.3.1), and magrittr (v. 2.0.3).

For manuscripts utilizing custom algorithms or software that are central to the research but not yet described in published literature, software must be made available to editors and reviewers. We strongly encourage code deposition in a community repository (e.g. GitHub). See the Nature Portfolio guidelines for submitting code & software for further information.

# Data

Policy information about availability of data

All manuscripts must include a data availability statement. This statement should provide the following information, where applicable:

- Accession codes, unique identifiers, or web links for publicly available datasets
- A description of any restrictions on data availability
- For clinical datasets or third party data, please ensure that the statement adheres to our policy

Project MinE data are available here: https://www.projectmine.com/research/data-sharing/. dbGAP datasets used are available under the following accession numbers: ALS compute (phs003184); Alzheimer's Disease Sequencing Project (ADSP) (phs000572); Autism Sequencing Consortium (ASC) (phs000298); Sweden-Schizophrenia Population-Based Case-Control Exome Sequencing (phs000473); Inflammatory Bowel Disease Exome Sequencing Study (phs001076); Myocardial Infarction Genetics Exome Sequencing Consortium: Ottawa Heart Study (phs000806); Myocardial Infarction Genetics Exome Sequencing Consortium: Malmo Diet and Cancer Study (phs001101); Myocardial Infarction Genetics Exome Sequencing Consortium: U. of Leicester (phs001000); Myocardial Infarction Genetics Exome Sequencing Consortium: Italian Atherosclerosis Thrombosis and Vascular Biology (phs000814); NHLBI GO-ESP: Women's Health Initiative Exome Sequencing Project (WHI) - WHISP (phs000281); Building on GWAS for NHLBI diseases: The US CHARGE Consortium (CHARGE-S): CHS (phs000667); Building on GWAS for NHLBI Diseases: the US CHARGE Consortium (CHARGE-S): ARIC (phs000668); Building on GWAS for NHLBI diseases: the US CHARGE consortium (CHARGE-S): FHS (phs000651); NHLBI GO-ESP Family Studies: Idiopathic Bronchiectasis (phs000518); NHLBI GO-ESP: Family Studies (Hematological Cancers) (phs000632); NHLBI GO-ESP: Family Studies: (familial atrial fibrillation) (phs000362); NHLBI GO-ESP: Heart Cohorts Exome Sequencing Project (ARIC) (phs000398); NHLBI GO-ESP: Heart Cohorts Exome Sequencing Project (CHS) (phs000400); NHLBI GO-ESP: Heart Cohorts Exome Sequencing Project (FHS) (phs000401); NHLBI GO-ESP: Lung Cohorts Exome Sequencing Project (asthma) (phs000422); NHLBI GO-ESP: Lung Cohorts Exome Sequencing Project (COPDGene) (phs000296); GO-ESP: Family Studies (Thoracic aortic aneurysms leading to acute aortic dissections) (phs000347). NHLBI TOPMed: Genomic Activities such as Whole Genome Sequencing and Related Phenotypes in the Framingham Heart Study (phs000974); NHLBI TOPMed: Genetics of Cardiometabolic Health in the Amish (phs000956); NHLBI TOPMed: Genetic Epidemiology of COPD (COPDGene) (phs000951); NHLBI TOPMed: The Vanderbilt Atrial Fibrillation Registry (VU_AF) (phs001032); NHLBI TOPMed: Cleveland Clinic Atrial Fibrillation (CCAF) Study (phs001189); NHLBI TOPMed: Partners HealthCare Biobank (phs001024); NHLBI TOPMed - NHGRI CCDG: Massachusetts General Hospital (MGH) Atrial Fibrillation Study (phs001062); NHLBI TOPMed: Novel Risk Factors for the Development of Atrial Fibrillation in Women (phs001040); NHLBI TOPMed - NHGRI CCDG: The Vanderbilt AF Ablation Registry (phs000997); NHLBI TOPMed: Heart and Vascular Health Study (HVH) (phs000993); NHLBI TOPMed - NHGRI CCDG: Atherosclerosis Risk in Communities (ARIC) (phs001211); NHLBI TOPMed: The Genetics and Epidemiology of Asthma in Barbados (phs001143); NHLBI TOPMed: Women's Health Initiative (WHI) (phs001237); NHLBI TOPMed: Whole Genome Sequencing of Venous Thromboembolism (WGS of VTE) (phs001402); NHLBI TOPMed: Trans-Omics for Precision Medicine (TOPMed) Whole Genome Sequencing Project: Cardiovascular Health Study (phs001368). All participants gave written informed consent, and all studies were approved by the institutional review boards of the respective participating centers.

# Research involving human participants, their data, or biological material

Policy information about studies with human participants or human data. See also policy information about sex, gender (identity/presentation), and sexual orientation and race, ethnicity and racism.

| | |
|---|---|
| Reporting on sex and gender | Sex was included as a covariate in all analyses that included individual-level data (single variant analyses; gene, domain, and gene set-based ultra-rare burden analyses). Self-reported sex was used if available, otherwise genetically inferred sex (plink software) was used. |
| Reporting on race, ethnicity, or other socially relevant groupings | No socially constructed or socially relevant categorization variables, such as self-reported race or ethnicity, were used for participant categorization or analysis in this manuscript. Genetic ancestry was inferred for all participants using the LASER software (Supplementary Figs. 2 and 11), and individuals of predominantly European ancestry were retained to ensure coverage of both cases and controls across ancestry space. The first ten principal components derived from common variants (MAF > 0.01) were included as covariates in all presented analyses that included individual-level data (single variant analyses; gene, domain, and gene set-based ultra-rare burden analyses). |
| Population characteristics | The discovery cohort (post-QC) included 13,138 patients with ALS (7,836 male, 5,302 female) and 69,775 control subjects (34,778 male and, 34,997 female)<br>The replication cohort (post-QC) included 4,781 patients with ALS (2,833 male, 1,948 female) and 130,928 control subjects (58,516 male and 72,412 female). |
| Recruitment | The discovery cohort included 15,862 patients with ALS and 78,683 control subjects, totaling 94,545 subjects of which 21,102 were subjected to whole-genome sequencing (WGS) and 73,443 to whole-exome sequencing (WXS). Case cohorts included the Project MinE ALS sequencing consortium (7,614 cases, 2,605 controls), the NYGC ALS Consortium (2,650 cases, 342 controls), the ALS Sequencing Consortium (2,851 cases), two cohorts from the FALS consortium (1,277 cases; phs001585), the NIH Exome Sequencing of FALS Project (194 cases; phs000101) two Australian cohorts described in (Garton et al., 2017) (125 cases and 18 controls) and (McCann et al., 2021) (568 cases), and a Chinese MND cohort described in (Gratten et al., 2017) (583 cases, 182 controls). All patients were diagnosed with definite, probable, or probable laboratory-supported ALS according to the revised El Escorial Criteria. Control cohorts included 7,323 samples from the NHLBI TOPMed research programme, 49,981 samples from the UK Biobank, and 18,232 samples across 7 cohorts from dbGAP.<br><br>The replication cohort included 5,404 patients with ALS and 133,823 control subjects, totaling 139,227 subjects, of which all were subjected to WGS. Cohorts include the Project MinE ALS sequencing consortium (1,510 cases, 169 controls), the NYGC ALS consortium (1,257 cases, 69 controls), ALS compute (1,870 cases, 1,820 controls; phs003184) and the UK Biobank (767 cases, 131,765 controls). During sample quality control, individuals who were duplicates or related up to the second degree to any participant in the discovery cohort were excluded. |
| Ethics oversight | This study was approved by the institutional review boards of all participating centers, written informed consent for research |

| Ethics oversight | was obtained from each individual, and the study was approved by the Medical Ethical Testing Committee NedMec and the Biobanks Testing Committee of UMC Utrecht. |

Note that full information on the approval of the study protocol must also be provided in the manuscript.

# Field-specific reporting

Please select the one below that is the best fit for your research. If you are not sure, read the appropriate sections before making your selection.

☒ Life sciences  ☐ Behavioural & social sciences  ☐ Ecological, evolutionary & environmental sciences

For a reference copy of the document with all sections, see nature.com/documents/nr-reporting-summary-flat.pdf

# Life sciences study design

All studies must disclose on these points even when the disclosure is negative.

| Sample size | The discovery dataset included 15,862 patients with ALS and 78,683 control subjects. After quality control and relatedness filtering the discovery analysis cohort included 13,138 patients with ALS and 69,775 controls. The sample size was not predetermined; instead, we assembled the largest possible sample size to optimize power. In the manuscript we show that we were adequately powered to detect most of the known ALS genes, including ultra-rare variants that have hitherto not been detected in case-control studies. The replication cohort included 5,404 patients with ALS and 133,823 control subjects. After quality control and relatedness filtering the replication analysis cohort included 4,781 patients with ALS and 130,928 controls. Power analyses showed that this sample size provides between 32¬--91% statistical power for replication across candidate variants and genes. |
| Data exclusions | Samples and variants were subjected to stringent quality control as detailed in the Methods section. Moreover, we retained only unrelated individuals (>= 2nd degree), and variants were included when predicted to have a moderate- or high-impact as predicted by the snpEff software and were either low-frequency (MAF < 0.05) in the single variant analyses or ultra-rare (<=5 carriers) in the burden analyses. |
| Replication | For replication, we generated a cohort comprising 4,781 patients with ALS and 130,928 controls after applying stringent quality control criteria identical to those used in the discovery set. This confirmed the single variant association in YKT6 (p.Y64C) at replication-wide significance. In addition, the single variant associations in HTR3C (p.T186A) KNTC1 (p.W287R), and GBGT1(p.R152L) meta-analysis of discovery and replication cohorts showed greater significance than the discovery phase alone. TTC3, KIF4A, UNC13C and CAPN2 could not be replicated with available samples and are thus explicitly presented as candidates for future research that only achieve significance in the discovery phase. |
| Randomization | This study used a case-control design, and therefore randomization is not applicable. All analyses that included individual-level data were adjusted for sex, ten principal components and the total number of synonymous variants in each individual in order to account for potential confounding of sex, ancestry and technical factors. |
| Blinding | This study used a case-control design, and therefore blinding is not applicable. |

# Reporting for specific materials, systems and methods

We require information from authors about some types of materials, experimental systems and methods used in many studies. Here, indicate whether each material, system or method listed is relevant to your study. If you are not sure if a list item applies to your research, read the appropriate section before selecting a response.

## Materials & experimental systems

| n/a | Involved in the study |
|---|---|
| ☒ | ☐ Antibodies |
| ☒ | ☐ Eukaryotic cell lines |
| ☒ | ☐ Palaeontology and archaeology |
| ☒ | ☐ Animals and other organisms |
| ☒ | ☐ Clinical data |
| ☒ | ☐ Dual use research of concern |
| ☒ | ☐ Plants |

## Methods

| n/a | Involved in the study |
|---|---|
| ☒ | ☐ ChIP-seq |
| ☒ | ☐ Flow cytometry |
| ☒ | ☐ MRI-based neuroimaging |

## Plants

| | |
|---|---|
| Seed stocks | *Report on the source of all seed stocks or other plant material used. If applicable, state the seed stock centre and catalogue number. If plant specimens were collected from the field, describe the collection location, date and sampling procedures.* |
| Novel plant genotypes | *Describe the methods by which all novel plant genotypes were produced. This includes those generated by transgenic approaches, gene editing, chemical/radiation-based mutagenesis and hybridization. For transgenic lines, describe the transformation method, the number of independent lines analyzed and the generation upon which experiments were performed. For gene-edited lines, describe the editor used, the endogenous sequence targeted for editing, the targeting guide RNA sequence (if applicable) and how the editor was applied.* |
| Authentication | *Describe any authentication procedures for each seed stock used or novel genotype generated. Describe any experiments used to assess the effect of a mutation and, where applicable, how potential secondary effects (e.g. second site T-DNA insertions, mosiacism, off-target gene editing) were examined.* |

