## [Peer Review File · Nature Genetics]

Large-scale exome analyses reveal novel rare variant contributions in amyotrophic lateral sclerosis

Corresponding Author: Professor Jan Veldink

This manuscript has been previously reviewed at another journal. This document only contains information relating to versions considered at Nature Genetics.

Version 0:

Decision Letter:

Our ref: NG-A70031-T

26th September 2025

Dear Jan,

Your revised manuscript "Large-scale exome analysis reveals novel rare variant contributions in ALS" (NG-A70031-T) has been seen by the original Nature referees. As you will see from their comments below, they find that the paper has improved in revision, and therefore we will be happy in principle to publish it in Nature Genetics as an Article pending final revisions to comply with our editorial and formatting guidelines.

We are now performing detailed checks on your paper, and we will send you a checklist detailing our editorial and formatting requirements soon. Please do not upload the final materials or make any revisions until you receive this additional information from us.

Thank you again for your interest in Nature Genetics. Please do not hesitate to contact me if you have any questions.

Sincerely,
Kyle

Kyle Vogan, PhD
Senior Editor
Nature Genetics
<https://orcid.org/0000-0001-9565-9665>

Reviewer #1 (Remarks to the Author):

The manuscript; Large-scale exome analysis reveals novel rare variant contributions to ALS and identifies YKT6 and ARPP21 as risk genes by Hop et al. analyzes a large exome sequencing dataset comprising of 13,138 patients with ALS and 69,775 controls, with independent replication in 4,781 patients with ALS and 130,928 controls of European descent (after filtering). The rationale for the approach is that in ALS rare variants with low-to-moderate penetrance seem to contribute to a significant proportion of genetic risk. The authors identify a number of genes (both known and novel genes) that confer risk to onset and/or progression of disease.

Overall the manuscript is clearly written. The quality control measures are well thought through and the different analyses are well executed. The filtering of samples and choice of analysis parameters are clearly described.

The results are convincing and in line with earlier observations that rare variants both with high and low-to-moderate penetrance play an important role in the genetics of ALS.

The functions of the identified novel genes fit well into what we currently know of the pathogenesis of ALS and provide more resolution into the affected molecular pathways.

Given that many identified variants have a low-moderate effect size with a small odds ratio, it would be of interest to see if cases carry multiple damaging variants and what the odds ratio of the total genetic risk (Polygenic Risk Score) would be compared to controls. And if there is a clustering of damaging variants in specific pathways for different groups of patients. Dissecting potential oligogenic contributions to ALS is of great interest, and to this end the authors have conducted a series of new analyses to address this and found that carrying multiple damaging variants cumulatively increases risk of ALS with Odds Ratios increasing with more contributing variants up to 4.26. Although rare even for definite ALS genes additional damaging variants are identified. It would be of interest to see if this has an effect on the clinical phenotype or progression rate of the disease.

The study is limited to independent (less than 2nd degree) patients of European ancestry after filtering for ancestry. In the future it would be of interest to see data on a more diverse population.

In addition, even in the European population, significant differences in allele frequencies are detected in different geographical regions. It would be informative to determine if individuals carrying ultra rare variants come from a specific geographical region and assess allele frequencies in closely matched controls. The authors have performed a more detailed analysis taking this point into account but in many of the "sub" cohorts currently lack the power. Thus, at this point this question cannot yet be addressed.

Importantly for understanding the genetic risk and for designing follow up wet lab experiments the authors have included (when possible) information on the directionality of identified variants or their functional consequences. This is a start to help understand the precise mechanisms by which these variants could contribute to ALS.

Reviewer #2 (Remarks to the Author):

This revised manuscript by Hop and colleagues presents a very comprehensive genetic analysis of the contribution of rare variants to ALS. It has much improved from the original submission and my concerns have all been addressed. In particular the addition of the new analyses focused on the oligogenic nature of ALS where the presence of having multiple variants is analyzed provides additional important data. The sensitivity analysis where it was shown that associations remained when potential overlapping samples from previous studies were removed was also important, as was the new structure in the manuscript where it is clearer now what was previously known and what is novel. The shift in emphasis away from the genes with less genetic emphasis also in the discussion is also an improvement.

Based on the specific answers to my previous comments, I just have one small remaining comment. I would suggest adding to the manuscript in the methods or results that C9orf72 status was not available for all cases and controls and provide the specific numbers. The authors now added to the methods that "Patients were included in this study irrespective of their carrier status for variants in known ALS genes." This is useful. However, the following information would also be of interest to include "In total, we have data on C9orf72 carrier status for 8,610 patients (~2/3) and 2,641 controls. Among these, 8% of patients and 0.5% of controls carry the repeat expansion." Given that there is a lot of reference to C9orf72 in the manuscript, also in combination with other mutations, it is important to realize the actual frequency of co-occurrence of C9orf72 with other mutations could be even higher.

Reviewer #3 (Remarks to the Author):

Authors have been responsive to my prior review. The identified ALS-associated genes are either novel or reinforce data for prior "lower confidence" genes.

I maintain that the work is an immense effort that has provided new insights into future pathways of study for ALS pathogenesis and potentially offers ultra-rare variant carriers a path towards personalized medicine.

Importantly, the text has been extensively revised to more accurately reflect the data presented.

Reviewer #4 (Remarks to the Author):

This review was co-authored by an established researcher and an early-career researcher, as part of Nature Genetics' initiative to support peer-review training and ensure appropriate recognition for early-career scientists.

In their revised manuscript, the authors present extensive additional analyses, clarify their key findings, and sharpen the focus of their main message. They substantially rewrote the results and discussion sections, effectively addressing all reviewers' comments. We fully support the publication of this version in Nature Genetics and commend the authors for their timely and thorough revisions.

Reviewer #5 (Remarks to the Author):

This review was co-authored by an established researcher and an early-career researcher, as part of Nature Genetics' initiative to support peer-review training and ensure appropriate recognition for early-career scientists.

In their revised manuscript, the authors present extensive additional analyses, clarify their key findings, and sharpen the focus of their main message. They substantially rewrote the results and discussion sections, effectively addressing all reviewers' comments. We fully support the publication of this version in Nature Genetics and commend the authors for their timely and thorough revisions.

Reviewer #1 (Remarks to the Author):

The manuscript; Large-scale exome analysis reveals novel rare variant contributions to ALS and identifies YKT6 and ARPP21 as risk genes by Hop et al. analyzes a large exome sequencing dataset comprising of 13,138 patients with ALS and 69,775 controls, with independent replication in 4,781 patients with ALS and 130,928 controls of European descent (after filtering). The rationale for the approach is that in ALS rare variants with low-to-moderate penetrance seem to contribute to a significant proportion of genetic risk. The authors identify a number of genes (both known and novel genes) that confer risk to onset and/or progression of disease.

Overall the manuscript is clearly written. The quality control measures are well thought through and the different analyses are well executed. The filtering of samples and choice of analysis parameters are clearly described.

The results are convincing and in line with earlier observations that rare variants both with high and low-to-moderate penetrance play an important role in the genetics of ALS.

The functions of the identified novel genes fit well into what we currently know of the pathogenesis of ALS and provide more resolution into the affected molecular pathways.

Given that many identified variants have a low-moderate effect size with a small odds ratio, it would be of interest to see if cases carry multiple damaging variants and what the odds ratio of the total genetic risk (Polygenic Risk Score) would be compared to controls. And if there is a clustering of damaging variants in specific pathways for different groups of patients.

Dissecting potential oligogenic contributions to ALS is of great interest, and to this end the authors have conducted a series of new analyses to address this and found that carrying multiple damaging variants cumulatively increases risk of ALS with Odds Ratios increasing with more contributing variants up to 4.26. Although rare even for definite ALS genes additional damaging variants are identified. It would be of interest to see if this has an effect on the clinical phenotype or progression rate of the disease.

We agree with the reviewer that it is of interest. We have therefore performed cumulative risk analyses for age at onset and survival and have added these to the manuscript (Rebuttal figure 1 below; lines 297-298, Supplementary Figure 8).

In short, we did not find a clear dose-response relationship between the number of low-frequency qualifying variants ($MAF < 0.05$; moderate- or high impact) they carried across definitive ALS genes. Importantly, such an association was also not observed when carrying a single variant compared to non-carriers. This result is not entirely unexpected, as many variants/genes associated with ALS risk don't have discernable effects on clinical phenotype. A further explanation is that our analysis aggregates all variants without considering their direction of effect. For example, *SOD1* p.D91A is associated with longer survival whereas *SOD1* A5V is associated with shorter survival;

aggregating opposing effects can mask true signal. Therefore, future analyses accounting for directionality are recommended to explore this further, which we considered beyond the scope of the current study.

Rebuttal figure 1. Cumulative burden analyses of age of onset and survival. Cumulative burden of carrying multiple high- or moderate-impact variants among Definitive ALS genes as curated by the GCEP. **a**, age of onset: shown are the effect estimates in years (center) and 95% confidence intervals (CI; error bars) (x-axis), stratified by the number of risk variants carried (y-axis). **b**, survival: shown are the log-transformed hazard ratios (center) and 95% confidence intervals (error bars) stratified by the number of risk variants carried (y-axis).

The study is limited to independent (less than 2nd degree) patients of European ancestry after filtering for ancestry. In the future it would be of interest to see data on a more diverse population.

In addition, even in the European population, significant differences in allele frequencies are detected in different geographical regions. It would be informative to determine if individuals carrying ultra rare variants come from a specific geographical region and assess allele frequencies in closely matched controls. The authors have performed a more detailed analysis taking this point into account but in many of the "sub" cohorts currently lack the power. Thus, at this point this question cannot yet be addressed.

Importantly for understanding the genetic risk and for designing follow up wet lab experiments the authors have included (when possible) information on the directionality of identified variants or their functional consequences. This is a start to help understand the precise mechanisms by which these variants could contribute to ALS.

We agree with the reviewer that follow up wet lab experiments are key to understanding the mechanisms through which these variants could contribute to ALS and we agree that the directionality and predicted functional consequences can provide a starting point.

Reviewer #2 (Remarks to the Author):

This revised manuscript by Hop and colleagues presents a very comprehensive genetic analysis of the contribution of rare variants to ALS. It has much improved from the original submission and my concerns have all been addressed. In particular the addition of the new analyses focused on the oligogenic nature of ALS where the presence of having multiple variants is analyzed provides additional important data. The sensitivity analysis where it was shown that associations remained when potential overlapping samples from previous studies were removed was also important, as was the new structure in the manuscript where it is clearer now what was previously known and what is novel. The shift in emphasis away from the genes with less genetic emphasis also in the discussion is also an improvement.

We thank the reviewer for their support and for confirming that the revisions have significantly improved the quality and clarity of the manuscript.

Based on the specific answers to my previous comments, I just have one small remaining comment. I would suggest adding to the manuscript in the methods or results that C9orf72 status was not available for all cases and controls and provide the specific numbers. The authors now added to the methods that “Patients were included in this study irrespective of their carrier status for variants in known ALS genes.” This is useful. However, the following information would also be of interest to include “In total, we have data on C9orf72 carrier status for 8,610 patients (~2/3) and 2,641 controls. Among these, 8% of patients and 0.5% of controls carry the repeat expansion.” Given that there is a lot of reference to C9orf72 in the manuscript, also in combination with other mutations, it is important to realize the actual frequency of co-occurrence of C9orf72 with other mutations could be even higher.

We thank the reviewer for pointing this out. We agree this is an important point and have now clarified the missingness in the relevant sections in the manuscript (Methods: line 993, Extended Data Table 3, and Extended Data Fig. 5).

To clarify, all co-occurrence frequencies were calculated on a pairwise complete bases, i.e. for any given pair the analysis was restricted to the subset of individuals for whom data was non-missing for both. Therefore, while having complete C9 data for all patients would increase the total number of observed co-occurrences, we do not expect the frequency of co-occurrence to change substantially. To make this clear in the manuscript, we have now added the following statement to the Methods section (lines 996-997).

“For each variant pair only individuals with non-missing genotypes for both variants were included.”.

Reviewer #3 (Remarks to the Author):

Authors have been responsive to my prior review. The identified ALS-associated genes are either novel or reinforce data for prior "lower confidence" genes.

I maintain that the work is an immense effort that has provided new insights into future pathways of study for ALS pathogenesis and potentially offers ultra-rare variant carriers a path towards personalized medicine.

Importantly, the text has been extensively revised to more accurately reflect the data presented.

We thank the reviewer for their endorsement of our findings and for highlighting the value of this study.

Reviewer #4 (Remarks to the Author):

This review was co-authored by an established researcher and an early-career researcher, as part of Nature Genetics' initiative to support peer-review training and ensure appropriate recognition

In their revised manuscript, the authors present extensive additional analyses, clarify their key findings, and sharpen the focus of their main message. They substantially rewrote the results and discussion sections, effectively addressing all reviewers' comments. We fully support the publication of this version in Nature Genetics and commend the authors for their timely and thorough revisions.

We thank the reviewers for their positive assessment and their support for the publication of our work.

Reviewer #5 (Remarks to the Author):

This review was co-authored by an established researcher and an early-career researcher, as part of Nature Genetics' initiative to support peer-review training and ensure appropriate rec

In their revised manuscript, the authors present extensive additional analyses, clarify their key findings, and sharpen the focus of their main message. They substantially rewrote the results and discussion sections, effectively addressing all reviewers'

comments. We fully support the publication of this version in Nature Genetics and commend the authors for their timely and thorough revisions.

We thank the reviewers for their positive assessment and their support for the publication of our work.